# Lipid droplets are a metabolic vulnerability in melanoma

Dianne Lumaquin-Yin [1,2], Emily Montal[1], Eleanor Johns [1], Arianna Baggiolini[3], Ting-Hsiang Huang[1], Yilun Ma[1,2], Charlotte LaPlante[1,2], Shruthy Suresh[1], Lorenz Studer [3] & Richard M. White [1,4] ✉

Melanoma exhibits numerous transcriptional cell states including neural crest-like cells as well as pigmented melanocytic cells. How these different cell states relate to distinct tumorigenic phenotypes remains unclear. Here, we use a zebrafish melanoma model to identify a transcriptional program linking the melanocytic cell state to a dependence on lipid droplets, the specialized organelle responsible for lipid storage. Single-cell RNA-sequencing of these tumors show a concordance between genes regulating pigmentation and those involved in lipid and oxidative metabolism. This state is conserved across human melanoma cell lines and patient tumors. This melanocytic state demonstrates increased fatty acid uptake, an increased number of lipid droplets, and dependence upon fatty acid oxidative metabolism. Genetic and pharmacologic suppression of lipid droplet production is sufficient to disrupt cell cycle progression and slow melanoma growth in vivo. Because the melanocytic cell state is linked to poor outcomes in patients, these data indicate a metabolic vulnerability in melanoma that depends on the lipid droplet organelle.

Phenotypic heterogeneity is influenced both by cell-intrinsic factors and tumor microenvironmental (TME) interactions[1–4]. In melanoma, both bulk and single-cell RNA-sequencing (scRNA-seq) has revealed numerous transcriptional cell states linked to distinct phenotypes[1–5]. For example, early studies demonstrated two unique gene signatures associated with proliferative versus invasive cell states[6,7]. These signatures have become increasingly nuanced, with most human melanoma exhibiting multiple transcriptional states linked to phenotypes such as metastasis, drug resistance, and immune evasion[2,3,8–10]. While these states are somewhat plastic, their maintenance across different patients suggests that each state may exhibit unique biological properties. One dimension of this phenotype spectrum is the degree of cellular differentiation. Melanocytes arise from the progressive differentiation of neural crest cells to melanoblasts, and finally differentiated melanocytes[11]. scRNA-seq has revealed that most melanomas contain a mixture of undifferentiated neural crest-like cells as well as more mature, pigmented melanocytic cells[2,4]. The less mature, neural crest-like states have been implicated in melanoma initiation and metastasis and have been well characterized[12,13]. In contrast, fewer studies have focused on the role of the pigmented, mature melanocytic cell state. Prior to terminal differentiation, these pigmented melanocytic cells remain highly proliferative, suggesting they may play a key role in tumor progression[9,14,15]. This state has translational importance since clinical data have correlated differentiated melanocytic identity with worse clinical outcomes[16–19]. Acquisition of a differentiated melanocytic cell identity has been associated with metastatic seeding and evasion of targeted therapy in patients, highlighting the need to uncover vulnerabilities in this cell state[19,20].

Dissecting the mechanisms driving this state requires tractable genetic models that recapitulate human melanoma phenotypic

[1]Department of Cancer Biology and Genetics, Memorial Sloan Kettering Cancer Center, New York, NY 10065, USA. [2]Weill Cornell/Rockefeller/Sloan-Kettering Tri-Institutional MD-PhD Program, New York, NY 10065, USA. [3]Center for Stem Cell Biology and Developmental Biology Program, Memorial Sloan Kettering Cancer Center, New York, NY 10065, USA. [4]University of Oxford, Ludwig Cancer Research, Nuffield Department of Medicine, Oxford, UK. ✉e-mail: richard.white@ludwig.ox.ac.uk

heterogeneity while preserving an immunocompetent TME. In this study, we use a zebrafish model of BRAF-driven melanoma coupled with analysis of human samples to uncover a metabolic dependency of the pigmented melanocytic state that is linked to lipid droplets.

## Results

### Melanocytic cell state upregulates oxidative metabolism in melanoma

We used our recently developed technique for generating melanoma in zebrafish, TEAZ (Transgene Electroporation of Adult Zebrafish)[21], to investigate phenotypic heterogeneity in a model of *BRAF^V600E p53^-/- PTEN^ko* melanoma (Fig. 1a). In this model, fish develop melanomas in fully immunocompetent animals at the site of electroporation in a median of 5 to 8 weeks. We performed scRNA-seq from $n = 6$ zebrafish tumors for a total of 3968 cells composed of melanoma and TME cells (Fig. 1b). After quality control and cluster annotation[22], we identified five melanoma clusters representing distinct transcriptional cell states (Fig. 1b, c and Supplementary Fig. 1a–e). Using differential gene expression and Gene Set Enrichment Analysis (GSEA), we identified the five melanoma clusters to represent transcriptional states of stressed, proliferative, invasive, inflammatory, and melanocytic identity (Fig. 1c, Supplementary Fig. 2a–d and Supplementary Data 1). The stressed cluster in our data set significantly enriched for genes like *gadd45ga*, *ddit3*, *fosb*, and *junba*, consistent with the stressed cellular state in previous zebrafish and human melanoma scRNA-seq profiling[2,8] (Fig. 1c and Supplementary Fig. 2a). Through GSEA, we found that the inflammatory cluster enriched for immune-related processes and the proliferative cluster strongly enriched for cell division processes (Fig. 1c and Supplementary Fig. 2b, c). Similarly, the invasive cluster enriched for genes like *krt18a.1* and *pdgfbb* expressed in previously characterized human melanoma invasive cell states[7], as well as cell migration processes (Fig. 1c and Supplementary Fig. 2d),

Differentiated melanocytes are marked by expression of pigmentation genes involved in melanin production[11,23]. The differentiated melanocytic cluster in our dataset showed enrichment for pigmentation genes expressed by differentiated melanocytes such as *tyrp1a*, *dct*, *pmela*, and *slc45a2* (Fig. 1c, d). To compare our zebrafish data to human data, we used module scoring to compare gene enrichment to the human melanocytic signature from ref. [4] and found the zebrafish melanocytic cluster to enrich for this differentiated melanocytic gene program (Fig. 1e). When we performed GSEA on the melanocytic cluster, we found enrichment for pigmentation related pathways among the top 10 GO Biological Processes as expected (Fig. 1f). Aside from this, we also noted that this cluster enriched for oxidative phosphorylation and fatty acid metabolism. To determine if this was unique to the zebrafish, we also analyzed the human melanoma brain and leptomeningeal metastases scRNA-seq dataset from ref. [24] (Supplementary Fig. 3a). We found that the cluster most significantly enriched for the Tsoi melanocytic gene program[4] also displayed enrichment for oxidative metabolism transcriptional programs (Supplementary Fig. 3b, c). This enrichment was not equally seen in other scRNA datasets[1,25], likely reflecting the small number of cells and varied biopsy sites seen in those studies. Altogether, within the available human datasets, our data suggest that acquisition of the melanocytic cell state correlates with a gene signature of oxidative metabolism.

To functionally test whether cells become more oxidative as they adopt a melanocytic cell state, we used a human pluripotent stem cell system. Either embryonic stem or induced pluripotent stem (iPS) cells can be differentiated into neural crest cells, melanoblasts, or mature melanocytes[11,13]. Similar to our zebrafish model, we introduced the *BRAF^V600E* oncogene and inactivated *PTEN* in the human pluripotent stem cells (Supplementary Fig. 4a, b). We then used the Seahorse Mito Stress Test to measure cellular OCR (oxidative consumption rate) and ECAR (extracellular acidification rate) in melanoblasts versus

melanocytes (Fig. 2a, b). These are markers for oxidative and glycolytic metabolism, respectively. We observed a robust increase in the basal OCR and OCR/ECAR ratio in the melanocytes compared to the melanoblasts, suggesting elevated oxidative metabolism in these more mature cells (Fig. 2b). Despite the advantage of the iPS system in terms of being isogenic at different states, one limitation is that the cells are analyzed at different timepoints (since melanoblasts emerge earlier than melanocytes). Thus to test this in a different way, we pharmacologically induced the melanocytic cell state in human A375 cells through increasing cAMP signaling via IBMX and Forskolin[26] (Fig. 2c). Treatment with IBMX and Forskolin resulted in upregulation of pigmentation genes, *dct* and *pmel*, and concurrent increases in the basal OCR and OCR/ECAR ratio (Fig. 2d, e). Collectively, these data provide evidence for oxidative metabolic rewiring in melanoma cells adopting a pigmented, melanocytic cell state.

### Fatty acids are fuel for oxidative metabolism in melanocytic cells

Oxidative metabolism can be fueled by various substrates including lipids, glucose, and glutamine[27]. Since we observed an enrichment for fatty acid pathways in the zebrafish melanocytic cluster, we next asked whether fatty acids were increasingly utilized as substrates for β-oxidation in the melanocytic cell state (Fig. 1f). We took advantage of the expansive melanoma data sets from the Cancer Cell Line Encyclopedia (CCLE)[28] and The Cancer Genome Atlas (TCGA)[29] which were recently characterized by gene expression to encompass undifferentiated/neural crest, transitory (an intermediate state), or melanocytic cell states[30]. As expected, these melanocytic samples demonstrate higher expression of pigmentation and melanocytic identity genes such as *MITF, DCT, PMEL*, and *TYR* (Supplementary Fig. 5a, c). These samples also upregulate *PPARGC1A* (Supplementary Fig. 5b, d), a gene important for driving mitochondrial oxidative metabolism and previously associated with *MITF* high melanomas[31,32]. To specifically assess fatty acid oxidation (FAO) gene expression, we scored enrichment for FAO gene signatures across cell states and found that melanocytic cell lines (Fig. 3a) and patient tumors (Fig. 3b) display the highest FAO gene score. These melanocytic samples upregulate the expression of key FAO genes such as *CPT2* and *HADHA*, although they downregulate *CPT1A* (Supplementary Fig. 5b, d).

To functionally test FAO, we performed the Seahorse FAO assay in human A375 melanoma cells ± IBMX or Forskolin to induce melanocytic differentiation in the presence of oleic acid (Fig. 3c, d and Supplementary Fig. 6a–e). We observed higher OCR with oleic acid supplementation in the melanocytic cells (compared to control A375 cells), suggesting that melanocytic cells have an increased ability to oxidize fatty acids (Fig. 3c and Supplementary Fig. 6d). Conversely in the presence of etomoxir, an inhibitor of FAO, the melanocytic cells demonstrated decreased OCR (Fig. 3d and Supplementary Fig. 6e). Although glucose and glutamine are available in the media to sustain oxidative phosphorylation, this suggests the melanocytic cells additionally elevate oxidative consumption of fatty acids.

FAO can be fueled by either mobilization of existing lipid stores or uptake from exogenous sources, and previous studies have shown that melanoma cells can upregulate exogenous fatty acid uptake through fatty acid transporter proteins[33,34]. To assess this in the melanocytic state, we added fluorescently labeled fatty acids to the media and measured fluorescence intensity as an indicator of fatty acid uptake into the cells[33,34]. This revealed that the more melanocytic cells had an increased number of lipid droplets (Supplementary Fig. 7), and a corresponding increase in lipid uptake compared to control A375 cells (Fig. 3e). However, they do not accelerate their uptake over time, suggesting that the number of lipid droplets induced by IBMX/Forskolin at that starting point (i.e., time 0 in Fig. 3e) is the main determinant of overall lipid uptake in these conditions. To further study this, we also assessed fatty acid uptake velocity[35] in BRAF^V600E

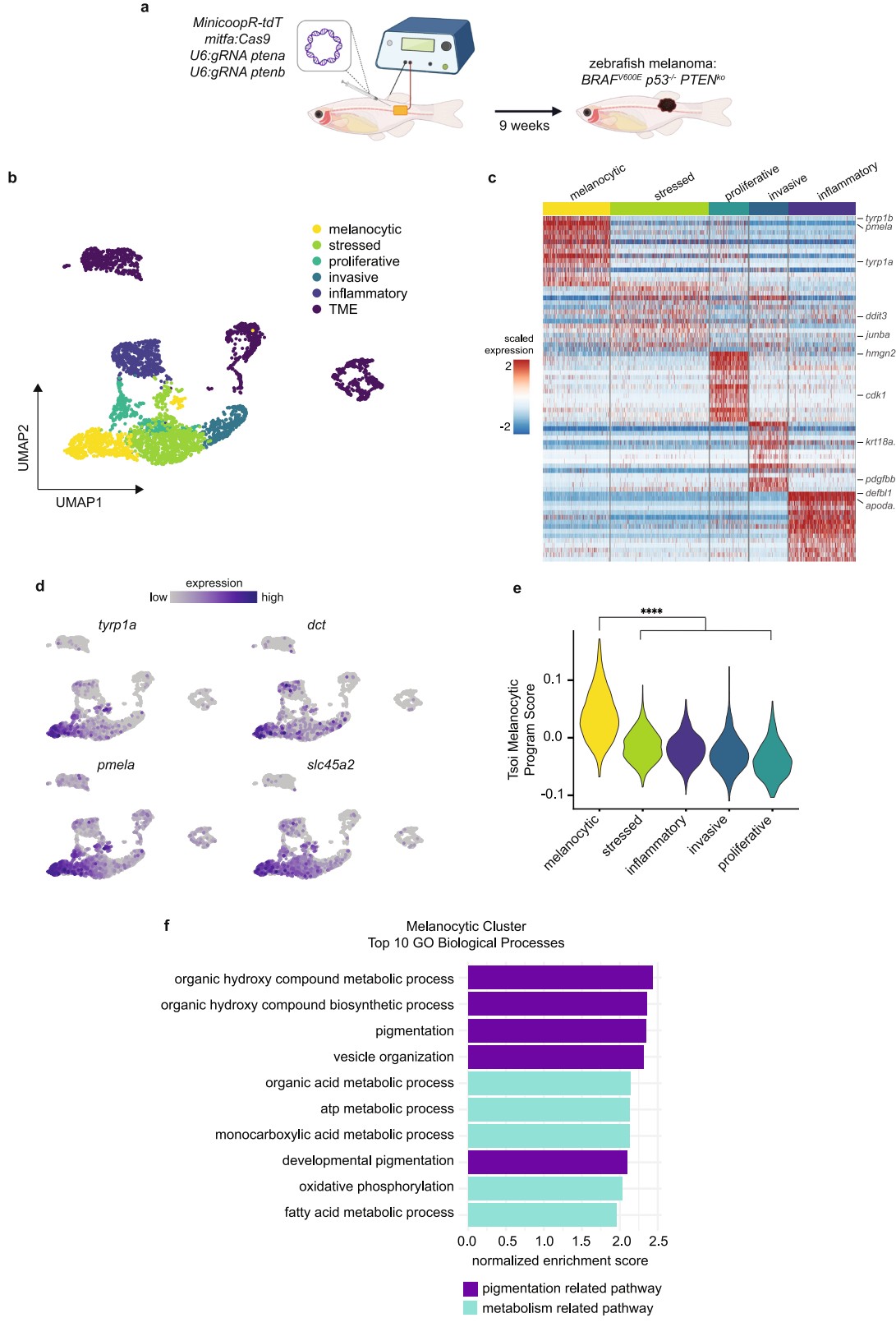

**Fig. 1 | Zebrafish melanoma displays distinct transcriptional states where melanocytic cell state upregulates oxidative metabolic pathways. a** Schematic of zebrafish TEAZ. Transgenic *casper;mitfa:BRAF^{V600E};p53^{-/-}* zebrafish were injected with tumor-initiating plasmids and electroporated to generate *BRAF^{V600E} p53^{-/-} PTEN^{ko}* melanomas. **b** UMAP dimensionality reduction plot of melanoma and TME cells. Cell assignments are labeled and colored. **c** Heatmap of top 15 differentially expressed genes in each melanoma transcriptional cell state. Select genes labeled.

**d** UMAP feature plots showing scaled gene expression of pigmentation genes (*tyrp1a, dct, pmela, slc45a2*). **e** Tsoi Melanocytic Program[4] module scores for cells in each melanoma cell state. Adjusted *p* values were calculated using Wilcoxon rank-sum test with Holm correction. **** *p* < 0.0001. **f** Top 10 enriched GO Biological Processes in melanocytic cluster filtered by multi-level Monte Carlo with Benjamini-Hochberg adjusted *p* values < 0.05. Pathways are color-coded based on pigmentation or metabolism-related pathways. Figure 1a was created with BioRender.com.

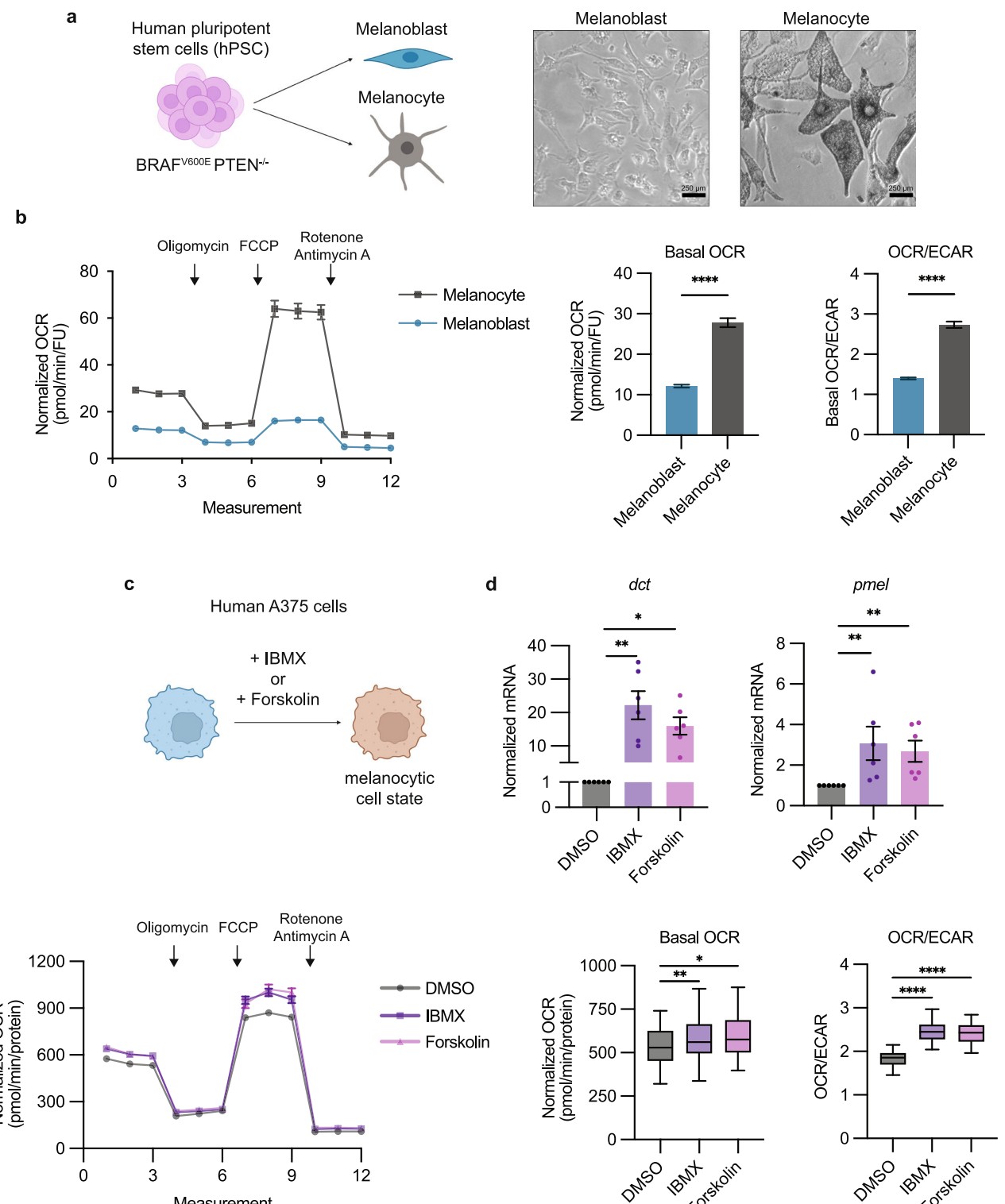

**Fig. 2 | Melanocytic cell state leads to oxidative metabolic rewiring. a** Schematic of hPSC differentiation to melanoblasts and melanocytes. Representative images for cell morphology of melanoblasts and melanocytes. **b** Normalized OCR measurements for melanoblasts and melanocytes. Basal OCR and OCR/ECAR values derived from measurement 3 (mean ± SEM, melanoblast $n = 111$ biologically independent replicates, IBMX $n = 108$ biologically independent replicates over $n = 3$ biologically independent experiments). Statistics via two-tailed t-test with Welch correction. **** $p < 0.0001$. **c** Schematic of inducing melanocytic cell state in human A375 cells via IBMX or Forskolin for 72 h. **d** qRT-PCR for melanocytic genes *dct* and *pmel* in human A375 cells (mean ± SEM, $n = 6$ biologically independent replicates over $n = 3$ biologically independent experiments). Statistics via Kruskal Wallis with Dunn's multiple comparisons test. * $p < 0.05$, ** $p < 0.01$. **e** Normalized OCR measurements for human A375 cells treated with DMSO, IBMX, or Forskolin. Basal OCR and OCR/ECAR values derived from measurement 3 (Tukey's boxplot center = median, box bounds = 25th and 75th percentile, whiskers = min and max, DMSO $n = 55$ biologically independent replicates, IBMX $n = 56$ biologically independent replicates, Forskolin n = 52 biologically independent replicates over $n = 3$ biologically independent experiments). Statistics via One-way ANOVA with Dunnett's multiple comparisons test. * $p = 0.016$, ** $p = 0.009$, **** $p < 0.0001$. Figure 2a, c was created with Biorender.com.

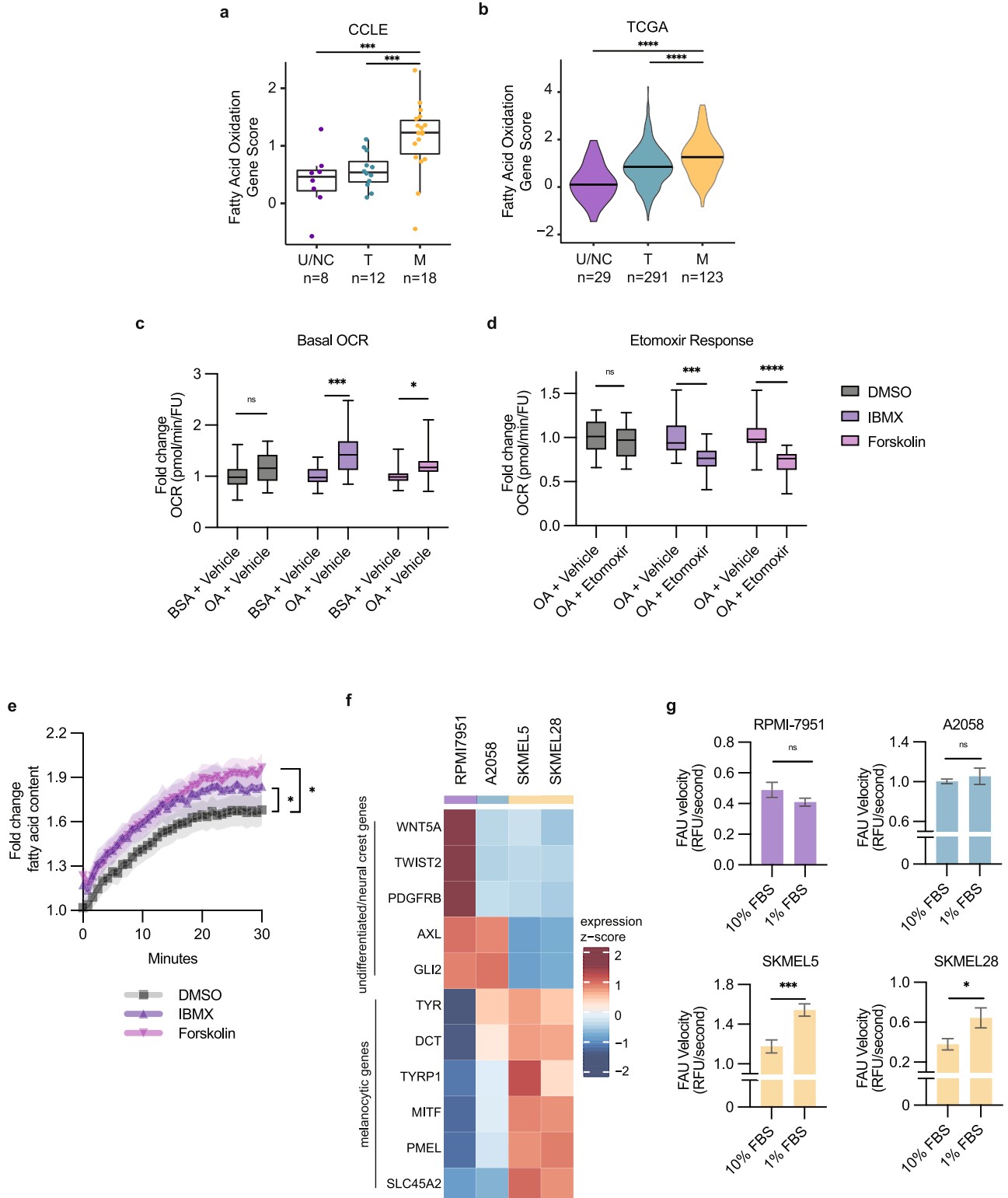

**Fig. 3 | Fatty acids are utilized by melanocytic cells for oxidative metabolism.** Fatty acid oxidation gene score for undifferentiated/neural crest (U/NC), transitory (T), and melanocytic (M) melanomas in the databases from: **a** CCLE (Tukey's boxplot center = median, box bounds = 25th and 75th percentile, whiskers = min and max, $n$ = number of independent cell lines) and **b** TCGA (Violin plot with bar indicating mean, $n$ = number of patient samples). Statistics via two-sided Wilcoxon rank sum test with Holm correction. *** $p < 0.001$, **** $p < 0.0001$. Fatty acid oxidation stress test in A375 cells showing fold change normalized OCR: **c** basal OCR from measurement 3 and **d** etomoxir response from measurement 9 (Tukey's boxplot center = median, box bounds = 25th and 75th percentile, whiskers = min and max,

$n$ = 3 biologically independent experiments). Statistics via two-tailed t-test with Holm-Sidak correction. * $p = 0.013$, *** $p < 0.001$, **** $p < 0.0001$. **e** Fold change in fatty acid uptake in drug-treated human A375 cells (mean ± SEM, $n$ = 3 biologically independent experiments). Statistics via area under the curve two-tailed t-test with Holm−Sidak correction. * $p < 0.05$. **f** Heatmap of CCLE melanocytic gene expression from RPMI-7951 (U/NC), A2058 (T), SKMEL5 (M), and SKMEL28 (M) cells. **g** Fatty acid uptake (FAU) velocity from RPMI-7951 (U/NC), A2058 (T), SKMEL5 (M), and SKMEL28 (M) cells (mean ± SEM, $n$ = 13 biologically independent replicates over $n$ = 3 biologically independent experiments). Statistics via two-tailed t-test. * $p = 0.03$, *** $p = 0.0006$.

melanoma cell lines identified as undifferentiated/neural crest (RPMI-7951), transitory (A2058), and melanocytic (SKMEL5 and SKMEL28) from the CCLE[30]. Correspondingly, the melanocytic cells demonstrate the highest expression of pigment production genes: *MITF, DCT, PMEL, TYR, TYRP1*, and *SLC45A2* (Fig. 3f). We compared how quickly cells take up exogenous fluorescently labeled fatty acids and found the melanocytic cell lines (SKMEL5 and SKMEL28) significantly increase fatty acid uptake velocity in lipid limited media (Fig. 3g). Altogether, this suggests melanocytic cells have an enhanced ability to uptake exogenous fatty acids and perform FAO.

### Melanocytic cells utilize lipid droplets to store fatty acids

While fatty acids can undergo metabolism through β-oxidation, excess levels of free fatty acids are toxic to cells and can limit proliferation, a phenomenon called lipotoxicity[36]. A major mechanism for avoiding such toxicity and maintaining proliferation is to package fatty acids as triacylglycerols in lipid storage organelles called lipid droplets[37]. Critical to this mechanism is DGAT1, an enzyme responsible for triacylglycerol synthesis[38]. When we looked at samples from the TCGA, we found that the melanocytic tumors express the highest *DGAT1* mRNA levels (Fig. 4a). We then tested the capacity of melanoma cell lines to generate triacylglycerols upon uptake of exogenous fatty acids such as oleic acid. As expected, all cell lines robustly increased triacylglycerol synthesis but to a higher degree in the melanocytic lines (SKMEL5 and SKMEL28) (Fig. 4b).

To regulate lipid availability, cells store fatty acids in the form of triacylglycerol within the lipid droplet which can shuttle fatty acids to the mitochondria for β-oxidation[39,40]. To test whether the more melanocytic cells utilized this mechanism of lipid storage, we added BODIPY conjugated fatty acid (BODIPY C16) and concurrently stained the cells with an antibody against PLIN2, a major lipid droplet protein that we and others have previously shown marks this organelle in melanoma cells[41–43]. Indeed we saw the BODIPY conjugated fatty acid colocalize with PLIN2, indicating fatty acid packaging into lipid droplets (Fig. 4c, d). To assess lipid droplet load, we quantified and found a positive correlation between BODIPY C16 and PLIN2 fluorescence area relative to cell area (Fig. 4e). Consistent with the imaging, the melanocytic cells display increased levels of BODIPY C16 and PLIN2 staining (Fig. 4e) and demonstrated a positive correlation between BODIPY C16 levels to melanocytic gene expression (Fig. 4f). Furthermore, we quantified the number of lipid droplets per cell in A375 cells after treatment with oleic acid (as a positive control) or after treatment with IBMX or Forskolin to induce the melanocytic state. This showed a significant increase in the number of lipid droplets in the melanocytic state compared to control cells (Supplementary Fig. 7a, b). Taken together, these data suggest that melanoma cells in the melanocytic state undergo metabolic rewiring to increase fatty acid conversion into triacylglycerol and upregulation of lipid droplet production.

### Loss of lipid droplets suppresses melanoma progression and disrupts cell cycle

Lipid droplet accumulation has been associated with increased melanoma cell proliferation and invasion leading to poor clinical outcomes[33,43]. Given the evidence linking lipid droplet accumulation to worse clinical outcomes, we next asked if disrupting lipid droplet formation would affect melanoma progression. To test this, we focused on DGAT1 since it is well known as a target to inhibit lipid droplet biogenesis[40,44]. More recently, DGAT1 has been linked to reducing oxidative stress and lipotoxicity in glioblastoma and melanoma[45,46]. To determine whether loss of DGAT1 would perturb lipid droplet formation in melanoma cells, we used a DGAT1 inhibitor and CRISPR/Cas9 to knock out *DGAT1* in our zebrafish melanoma lipid droplet reporter[41] (Fig. 5a and Supplementary Fig. 8a, b). Using imaging and flow cytometry, we found that pharmacologic inhibition or knockout of *DGAT1* suppressed lipid droplet biogenesis even when

challenged with exogenous fatty acid (Fig. 5a and Supplementary Fig. 8b). Next, we tested if disrupting lipid droplet biogenesis would affect melanoma progression in vivo by knocking out *dgat1a* in our TEAZ model of melanoma (Fig. 5b). Interestingly, knockout of *dgat1a* showed no difference in tumor size at early time points suggesting its loss has minimal effect in tumor initiation. In contrast, loss of *dgat1a* led to a significant reduction in tumor area at later time points (Fig. 5c). Overall, this suggests that lipid droplets play a role in later stages of tumor growth and progression.

Despite the reduction in tumor progression, the *dgat1a* knockout tumors still continued to grow, albeit at a reduced rate. Consistent with this, hematoxylin and eosin (H&E) and BRAF[V600E] staining showed both control and *dgat1a* knockout tumors could form advanced melanomas capable of tumor invasion beyond hypodermal layers like muscle (Fig. 5d and Supplementary Fig. 9a–c). To gain further insight into the mechanisms disrupting tumor progression in *dgat1* deficient tumors, we dissected and sorted tdTomato+ melanoma cells from control and *dgat1a* knockout tumors (Fig. 6a and Supplementary Data 2). Targeted sequencing of the *dgat1a* locus confirmed the presence of insertions and deletions (indel)[47] specifically in the *dgat1a* knockout tumors (Supplementary Fig. 8c). Furthermore, bulk RNA-sequencing of the control versus *dgat1a* knockout melanoma cells showed a significant reduction in *dgat1a* mRNA expression in the knockout tumors (Supplementary Fig. 8d).

We performed GSEA and observed that among the top 15 most significantly enriched pathways, the majority are negatively enriched cell cycle pathways. This suggested that inhibiting lipid droplet formation in melanoma cells results in reduced proliferative capacity, consistent with the smaller tumor size in *dgat1a* knockout tumors (Figs. 5c and 6b). Since targeted sequencing showed a mixture of wildtype and *dgat1a* knockout cells in the TEAZ melanomas (Supplementary Fig. 8c), we wondered whether wildtype cells compensated for the proliferative disadvantage of *dgat1a* knockout cells. As an orthogonal method to assess tumor proliferation, we performed a zebrafish melanoma transplant assay in the blastula and treated the animals with a pharmacologic DGAT1 inhibitor[8] (Fig. 6c). Using the blastula transplant assay, we found a striking reduction in tumor growth in animals treated with a DGAT1 inhibitor (Fig. 6d). To assess if this applies to human melanoma cells, we treated our panel of human melanoma cell lines with the DGAT1 inhibitor in nutrient limited media and measured proliferation. DGAT1 inhibition did not alter cell proliferation in the most undifferentiated cell line, RPMI-7951 (Fig. 6e). However, we observed reduced proliferation upon DGAT1 inhibition of more melanocytic cell lines (Fig. 6e). Altogether this suggests an increased sensitivity to lipid droplet suppression as cells acquire a melanocytic state, leading to reduced cell cycle progression.

We also considered the possibility that DGAT1 inhibition leads to glycolytic and oxidative metabolic rewiring. Recent work has shown that lipid droplets are necessary to preserve mitochondrial oxidative function especially during periods of nutrient stress[40,45]. To test whether lipid droplet loss disrupts glycolytic and mitochondrial oxidative function in melanoma cells, we inhibited DGAT1 in human melanoma cell lines in nutrient-limited media and performed the Seahorse ATP Rate Assay (Supplementary Fig. 10a–d). DGAT1 inhibition reduced ATP production from mitochondrial respiration in all cell lines though to a greater magnitude in the melanocytic lines (Supplementary Fig. 10e). There was a smaller effect on glycolytic metabolism with the exception of the A2058 which upregulated glycolysis (Supplementary Fig. 10f). Collectively, these findings suggest that lipid droplet inhibition in melanocytic cells can disrupt oxidative metabolism.

## Discussion

Phenotypic heterogeneity through non-genetic reprogramming is increasingly recognized as a mechanism for survival in tumors and a

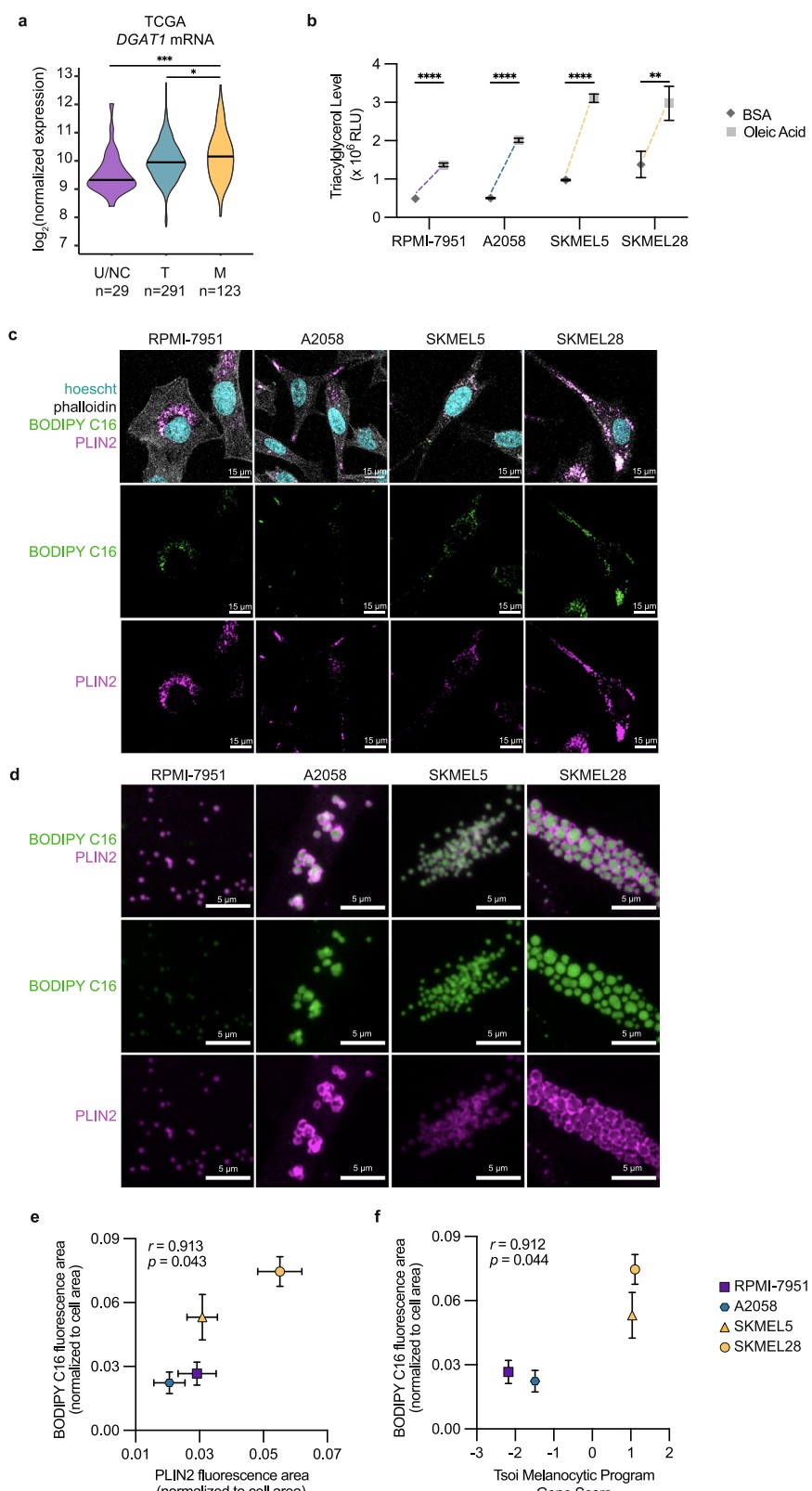

**Fig. 4 | Melanocytic cells increase lipid droplet production. a** TCGA $\log_2$ normalized mRNA expression for *DGAT1* (Violin plot with bar indicating mean). Statistics via two-sided Wilcoxon rank sum test with Holm correction. * $p = 0.03$, *** $p = 0.0002$. **b** Triacylglycerol levels in melanoma cell lines 24 h after addition of BSA or 100 µM oleic acid (mean ± SEM, $n = 12$ biologically independent replicates over $n = 3$ independent experiments). Statistics via two-tailed t-test with Holm-Sidak correction. ** $p = 0.009$, **** $p < 0.0001$. **c** Representative whole cell images of lipid droplets in melanoma cell lines 24 h after BODIPY C16 uptake ($n = 3$ biologically independent experiments). **d** Representative higher magnification images of lipid droplets in melanoma cell lines. Plots of (**e**) PLIN2 to BODIPY C16 fluorescence area and (**f**) Tsoi Melanocytic Program Gene Score to BODIPY C16 fluorescence area for melanoma cell lines (mean ± SEM, $n = 3$ biologically independent experiments). Statistics via Pearson correlation with one-tailed t-test.

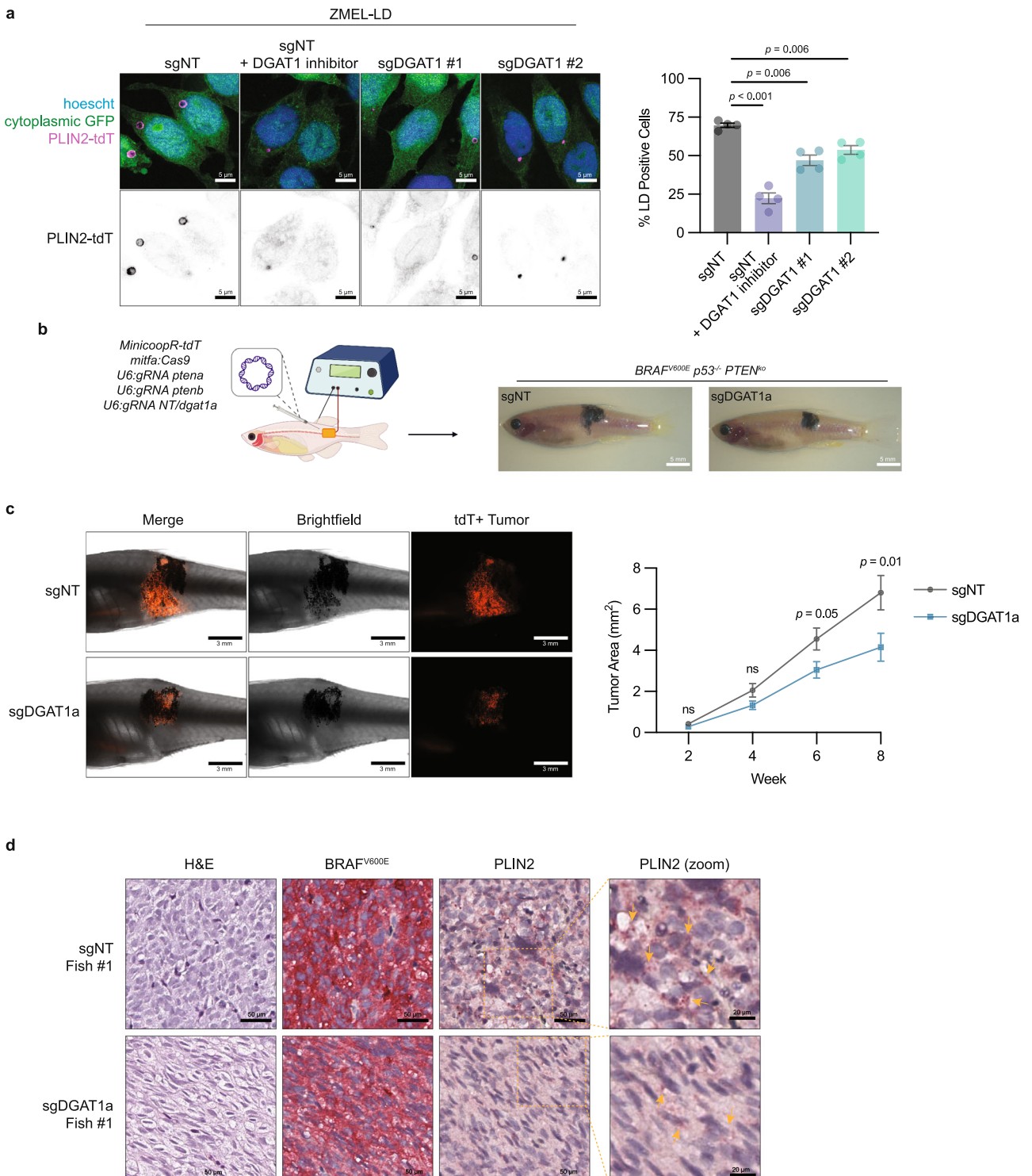

**Fig. 5 | Knockout of DGAT1 suppresses lipid droplet formation and tumor progression. a** Representative fluorescent images of ZMEL-LD lipid droplets marked by PLIN2-tdTOMATO. Cells were incubated with 100 μM oleic acid for 24 h and lipid droplets were quantified via flow cytometry (mean ± SEM, $n = 4$ biologically independent experiments). Statistics via two-tailed t-test with Holm-Sidak correction. **b** Schematic of zebrafish TEAZ. Transgenic *casper;mitfa:BRAF*[V600E]*;p53*[−/−] zebrafish were injected with tumor-initiating plasmids and electroporated to generate BRAF[V600E] p53[−/−] PTEN[ko] melanomas with normal or suppressed lipid droplet

formation. **c** Representative images of zebrafish flank with TEAZ-generated tumors. Corresponding quantification of tumor area via image analysis as described in Methods (mean ± SEM, $n = 3$ biologically independent injections, sgNT $n = 45$, sgDGAT1a $n = 49$). Statistics via two-sided Mann-Whitney U test at each time point. **d** Representative histological images from week 12 TEAZ generated tumors with H&E, BRAF[V600E], and PLIN2 staining ($n = 4$ fish per genotype). Yellow arrows denote punctate PLIN2 staining for lipid droplets. Figure 5b was created with Biorender.com.

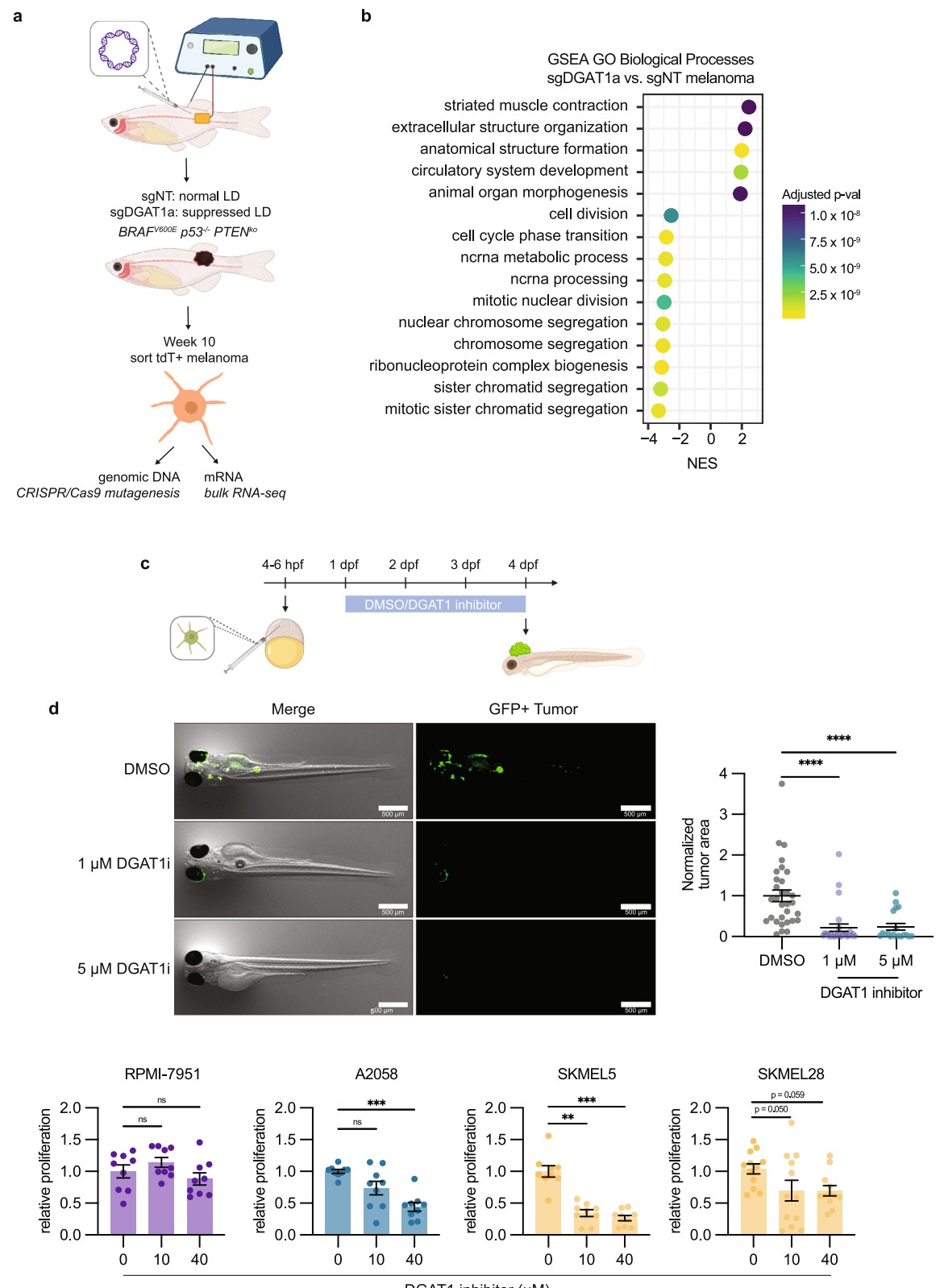

therapeutic barrier in melanoma[48]. One example of this reprogramming is metabolic rewiring in which cancer cells metabolically adapt to changing microenvironments and stressors[49]. Metabolic profiling across melanoma cell lines has shown that melanomas have the capacity to simultaneously perform glycolysis and oxidative metabolism even under stressors like hypoxia[50]. However, oxidative metabolism produces significantly more ATP than glycolysis, providing

necessary building blocks for cell growth and survival[50]. Previous reports have demonstrated that rewiring to increase oxidative metabolism can drive drug resistance and metastasis in melanoma[32,51,52]. One mechanism for this increased oxidative metabolic phenotype is mediated by the melanocyte master regulator, MITF, which regulates expression of the metabolic transcriptional coactivator, PGC1α[31]. Conversely, PGC1α can regulate MITF to induce melanogenesis[53]. Our

**Fig. 6 | Inhibiting lipid droplet formation suppresses tumor growth and cell cycle progression. a** Schematic of zebrafish TEAZ and sort for genomic DNA and mRNA. Transgenic *casper;mitfa:BRAF*$^{V600E}$*;p53*$^{-/-}$ zebrafish were injected with tumor-initiating plasmids and electroporated to generate *BRAF*$^{V600E}$ *p53*$^{-/-}$ *PTEN*$^{ko}$ melanomas with normal or suppressed lipid droplet formation. Melanoma cells were sorted to extract genomic DNA and mRNA. **b** Top 15 most significant (multi-level Monte Carlo by Benjamini-Hochberg adjusted *p* values) GO Biological Processes positively and negatively enriched in sgDGAT1a tumors. **c** Schematic of ZMEL-LD blastula transplant assay. **d** Representative images and tumor area quantification of blastula transplant. (mean ± SEM, DMSO *n* = 31 biologically independent replicates,

1 μM n = 27 biologically independent replicates, 5 μM *n* = 19 biologically independent replicates over *n* = 3 biologically independent experiments). Statistics via Kruskal Wallis with Dunn's multiple comparison test. **** *p* < 0.0001. **e** Relative cell proliferation in RPMI-7951 (U/NC), A2058 (T), SKMEL5 (M), and SKMEL28 (M) treated with indicated DGAT1i inhibitor concentrations in nutrient limited media. (mean ± SEM, RPMI-7951 *n* = 9 biologically independent replicates, A2058 *n* = 9 biologically independent replicates, SKMEL5 *n* = 9 biologically independent replicates, SKMEL28 *n* = 12 biologically independent replicates over *n* = 3 biologically independent experiments). Statistics via Kruskal Wallis with Dunn's multiple comparison test. ** *p* < 0.01, *** *p* < 0.001. Figure 6a, c was created with Biorender.com.

---

results are consistent with previous studies implicating a direct relationship between melanocytic identity and oxidative metabolism. However, the mechanisms for how melanoma cells determine which metabolic substrates to utilize for oxidative metabolism are not completely understood. The results of study show that fatty acids are one key substrate, and that this depends upon the uptake and processing of lipid droplets. While we cannot exclude that other substrates could be used, this data shows that fatty acids are especially important in the melanocytic state.

A critical aspect of cancer proliferation is acquisition of substrates to support the energetically expensive process of sustained cellular growth[49]. Many reports support the concept that melanoma cells actively scavenge lipids which can be utilized for processes like membrane formation and energy production[33,34,54,55]. Across melanoma cell lines and primary patient melanomas, we found that acquisition of melanocytic identity corresponds with enhanced fatty acid uptake and oxidation. Interestingly, a recent study found that the process of melanogenesis relies on elevated fatty acid oxidation in melanocytes and mouse B16 melanoma cells[56]. Our data support the idea that melanocytic identity and dependence on lipid metabolism are intrinsically coupled.

Despite the benefits of lipid uptake, lipid accumulation can lead to lipotoxicity. Thus, lipid droplets are critical for storing and facilitating fatty acid release when needed for biosynthetic or energetic purposes[39,40]. Here, we show that targeting DGAT1 impairs lipid droplet biogenesis consequently leading to suppressed tumor growth and metabolic dysfunction. Lipid droplet biogenesis through DGAT1 has been shown as essential to maintaining mitochondrial health and defending against lipid peroxidation in cancer[45,46]. Furthermore, melanoma resistance to therapy has been associated with dedifferentiation leading to increased sensitivity to lipid peroxidation and ferroptosis[4]. Thus it is possible that melanoma cells with enhanced lipid droplet capacity, such as those in a melanocytic cell state, are at a metabolic advantage. The pathways that allow some populations to overcome the metabolic stressors of lipid droplet depletion are still open questions.

Beyond cellular metabolism, lipid droplet accumulation is tied to cell fate as seen in neural stem and progenitor cells and colorectal cancer stem cells[57,58]. We found that lipid droplets increase with the acquisition of more differentiated transcriptional identity in melanoma cells. Altogether, this brings forward the question of how lipid droplets reflect cellular differentiation across different cell types. Recent evidence has shown that lipid droplets maintain physical contact with the mitochondria, endoplasmic reticulum, lysosome, Golgi apparatus, and peroxisome to function as an antioxidant organelle and mediate inter-organelle transport of macromolecules[37,59]. While our work focused on DGAT1, profiling of the lipid droplet proteome has revealed a surprisingly diverse breadth of metabolic, signaling, trafficking, and membrane organization proteins specifically embedded in the lipid droplet membrane[60]. Future studies will be needed to determine whether perturbing lipid droplet proteins disrupt communication between organelles during cellular demands of tumor progression. Our findings place lipid droplets at the crossroads of oxidative metabolism and lipid regulation, presenting an attractive

target at the intersection of metabolic processes necessary for sustained growth in melanoma.

## Methods

All studies comply with institutional ethics regulations. All zebrafish experiments are approved by MSKCC IACUC protocol number 12-05-008.

### Zebrafish husbandry

Zebrafish were housed in a temperature- (28.5 °C) and light-controlled (14 h on, 10 h off) room. Zebrafish were anesthetized using Tricaine (MS-222) with a stock of 4 g/l (protected from light) and diluted until the fish was immobilized. All procedures were approved by and adhered to Institutional Animal Care and Use Committee (IACUC) protocol #12-05-008 through Memorial Sloan Kettering Cancer Center.

### Cell culture

Human melanoma A375, RPMI-7951, A2058, SKMEL5, and SKMEL28 cells were obtained from ATCC and routinely confirmed to be free from mycoplasma. Cells were maintained in a 37 °C and 5% CO$_2$ humidified incubator. Cells were maintained in DMEM (Gibco, 11965) supplemented with 10% FBS (Gemini Bio, 100-500) and 1× penicillin/streptomycin/glutamine (Gibco, 10378016). Cells were used for experiments until passage 25.

Zebrafish ZMEL-LD cells were generated as recently described[41] and routinely confirmed to be free from mycoplasma. Cells were maintained in a 28 °C and 5% CO$_2$ humidified incubator. Cells were maintained in DMEM supplemented with 10% FBS, 1× penicillin/streptomycin/glutamine, and 1× GlutaMAX (Gibco, 35050061). Cells were used for experiments until passage 25.

### Zebrafish melanoma generated by TEAZ

Melanomas were generated by Transgene Electroporation in Adult Zebrafish as previously described[13,21]. To generate BRAF$^{V600E}$ p53$^{-/-}$ PTEN$^{ko}$ melanomas, adult (3–6 months) transgenic zebrafish (*casper (mitfa*$^{-/-}$*;mpv17*$^{-/-}$*);mitfa:BRAF*$^{V600E}$*;p53*$^{-/-}$) were injected with the following tumor initiating plasmids: MinicoopR-tdT (250 ng/μL), *mitfa:Cas9* (250 ng/μL), *U6:sgptena* (23 ng/μL), *U6:sgptenb* (23 ng/μL) and Tol2 (57 ng/μL). For *dgat1a* knockout experiments, zebrafish were injected with an additional guide plasmid of *U6:sgNT* (23 ng/μL) or *U6:sgdgat1a* (23 ng/μL). Adult male and female zebrafish were anesthetized in tricaine and injected with 1 μL of tumor-initiating plasmids below the dorsal fin, electroporated using the CM 830 Electro Square Porator from BTX Harvard Apparatus, and recovered in fresh water. For *dgat1a* knockout experiments, zebrafish were imaged every 2 weeks using brightfield and fluorescence imaging using a Zeiss AxioZoom V16 fluorescence microscope. To quantify tumor area, images were analyzed in MATLAB R2020a by quantifying pixels positive for melanin and tdTomato.

### Zebrafish sgRNA−target sequences
*U6:sgNT* - AACCTACGGGCTACGATACG
*U6:sgptena* - GAATAAGCGGAGGTACCAGG

*U6:sgptenb* - GAGACAGTGCCTATGTTCAG
*U6:sgdgat1a* - GTGACTCAAGCCAAACGCGG

## Zebrafish blastula transplant

ZMEL-LD cells were injected into the blastula of pre-epiboly *casper* embryos as previously described[8]. Briefly, 20 cells in 1 nL were injected into the animals using a quartz microneedle. Embryos were grown in E3 water for 24 h then dechorionated and placed in E3 with DMSO or 1 μM/5 μM DGAT1 inhibitor (T863, Cayman Chemical Company, 25807). E3 water with fresh drugs was changed every 24 h for a total of 72 h. Zebrafish were imaged using brightfield and fluorescent imaging using a Zeiss AxioZoom V16 fluorescence microscope to quantify tumor area in MATLAB 2020a[8].

## Zebrafish histology and immunohistochemistry

Zebrafish were sacrificed in an ice bath for at least 15 min. Zebrafish were fixed in 4% paraformaldehyde for 72 h at 4 °C, washed in 70% ethanol for 24 h, and then paraffin embedded. Fish were sectioned at 5 mm and placed on Apex Adhesive slides, baked at 60 °C, and then stained with hematoxylin & eosin or antibodies against BRAF^V600E (1:100, Abcam, ab228461) or PLIN2 (1:200, Proteintech, 15294-1-AP) using the BOND Polymer Refine Red Detection Kit (Leica Biosystems, DS9390). Whole slide scanning was performed on an Aperio AT2 digital whole slide scanner (Leica Biosystems). Histology was performed by Histowiz.

## scRNA-sequencing of zebrafish melanomas

Six zebrafish (3 male and 3 female) with melanoma (3 months post-TEAZ) were anesthetized and sacrificed using Tricaine. Tumor and adjacent tissue were dissected and dissociated using 0.16 mg/mL Liberase (Sigma-Aldrich, #5401020001) in 1x PBS and gentle pipetting with a wide-bore P1000 for 30 min at room temperature. Dissociation was terminated with addition of 250 μL FBS and samples were filtered through a 70 μm filter. Male and female zebrafish were labeled by sex using the 3' CellPlex Kit Set A (10x Genomics, 1000261) per the manufacturer's instructions. The resulting cell pellets were resuspended in 1x PBS and 1% UltraPure BSA (Thermo-Fisher, AM2616) and passed through a 40 μm filter into 5 ml polystyrene tubes for FACS. Cells were sorted to remove debris and doublets using BD FACSAria III cell sorter.

Library preparation and sequencing were done by the Single Cell Research Initiative and Integrated Genomics Organization at MSKCC. For cell encapsulation and library preparation, droplet-based scRNA-seq was performed on approximately 5900 cells using the Chromium Single Cell 3' Library and Gel Bead Kit v3 and Chromium Single Cell 3' Chip G (10x Genomics) into a single v3 reaction. GEM generation and library preparation were performed according to manufacturer instructions. Libraries were sequenced on a NovaSeq6000. Sequencing parameters: Read1 28 cycles, i5 10 cycles, i7 10 cycles, Read2 90 cycles. Sequencing depth was -51,000 reads per cell. Sequencing data were aligned to our reference zebrafish genome using CellRanger 6.0.2 (10x Genomics)[61].

Data were processed using R version 4.0.5 and Seurat version 4.0.2[22]. Cells with fewer than 200 unique genes and mitochondrial genes above 30% were filtered. Expression data were normalized using SCTransform with principal component analysis and UMAP dimensionality reduction performed at default parameters. Clustering was performed using the Seurat function FindClusters with a resolution of 0.4. Cluster annotation for zebrafish cell-type specific marker genes as done previously using FindAllMarkers[8,61].

Differentially expressed genes for pathway analysis were performed using the Seurat function FindMarkers comparing the tumor clusters. Ribosomal genes and genes with *p* values < 0.05 were filtered out. Ortholog mapping between zebrafish and humans was performed with DIOPT[13,61]. Gene set enrichment analysis was performed using fgsea 1.16.0 using the MSigDB GO biological processes (GO.db 3.12.1)[61].

The calculation of Tsoi Melanocytic Program[4] Score was determined using the Seurat AddModuleScore function with default parameters.

## Re-analysis of publicly available human melanoma RNA-seq

Smalley et al.[24] human melanoma scRNA-seq data were processed using R version 4.0.5 and Seurat version 4.0.2[22]. The counts' matrix was obtained from GEO (GSE1744401) and Seurat object was created with default parameters as described previously[61]. The calculation of Tsoi Melanocytic Program[4] Score was determined using the Seurat AddModuleScore function with default parameters.

Log$_2$ normalized counts from CCLE and TCGA melanoma datasets were retrieved from publicly available databases: https://depmap.org/portal/download/all/ and https://tcga.xenahubs.net. Fatty acid oxidation gene score was determined based on the Seurat AddModuleScore calculation using genes from GO: Fatty Acid Oxidation (GO:0019395).

## Bulk RNA-sequencing of zebrafish melanomas

Zebrafish tumors were dissected and sorted for tdTomato+ cells as described above. mRNA was extracted and prepared with SMARTer Universal Low Input RNA Kit for Sequencing (Takara) for 100 bp paired-end sequencing on the NovaSeq 6000 by the Integrated Genomics Organization at MSKCC. Sequencing reads underwent quality control with FASTQC 0.11.9, trimming with TRIMMOMATIC 14.0.1, and aligned using Salmon 1.4.0 to the *danio rerio* GRCz11.

Data analysis was conducted in R version 4.0.5. Differential expression was calculated using DESeq2 1.30.1 using default parameters. Significant differentially expressed genes were called if log$_2$ fold change > 1 and adjusted *p* value < 0.05. Ortholog mapping between zebrafish and human was performed with DIOPT[13,61]. Gene set enrichment analysis was performed using fgsea 1.16.0 using the MSigDB GO biological processes (GO.db 3.12.1)[61].

## Generation of PTEN knockout line

sgRNAs were cloned into the PX458 Cas9-GFP vector and introduced into dox-inducible BRAF^V600E hPSC by nucleofection as previously described[13]. This study was approved by the Tri-Institutional (MSKCC, Weill Cornell, Rockefeller University) Embryonic Stem Cell Research Oversight (ESCRO) Committee. Cells were FACS sorted 24 h post nucleofection, and individually seeded on a mouse embryonic fibroblast feeder layer in the presence of 10 mM ROCK-inhibitor in knockout serum replacement stem cell media[62] for 2 weeks. ROCK-inhibitor was removed from culture media after 4 days. Clones were transferred to vitronectin-coated plates and further maintained in E8. Full loss of PTEN expression was finally validated by Western blotting (anti-PTEN antibody, Cell Signaling, 9559S, 1:1000). sgRNAs come from ref. [63].

**Human PTEN sgRNA sequences.** sgRNA_F: CACCGAACTTGTCTTCCCGTCGTGT

sgRNA_R: AAACACACGACGGGAAGACAAGTTC

## Melanoblast differentiation protocol

Dual SMAD inhibition protocol was performed as previously described[11] and melanocytes differentiation was executed previously[13].

In brief, the dox-inducible BRAF^V600E PTEN KO hPSC was plated as a high-density monolayer with 150,000 cells per cm$^2$. This is a lower density than that one used for the WT cells, because of the higher proliferation rates due to the PTEN knockout. It is important to induce BRAF^V600E only at the end of the differentiation. This otherwise impairs hPSC differentiation.

Day −1: Plate hPSCs on Matrigel in E8 medium with 10 μM ROCKi (R&D, 1254).

Day 0−2: Change media every day with E6 media containing 1 ng/ml BMP4 + 10 μM SB + 600 nM CHIR.

Day 2−4: Change media with E6 media containing 10 μM SB + 1.5 μM CHIR.

Day 4–6: Change media with E6 media containing 1.5 µM CHIR.

Day 6–11: Change media every day with E6 media containing 1.5 µM CHIR + 5 ng/ml BMP4 100 nM EDN3.

## Flow cytometry-associated cell sorting

Dox-inducible BRAF$^{V600E}$ PTEN KO hPSC-derived melanoblasts were sorted at day 11 of differentiation using a BD-FACS Aria6 cell sorter at the Flow Cytometry Core Facility of MSKCC. The cells in differentiation were initially dissociated into single cells using Accutase (Innovative Cell Technologies, 397) for 20 min at 37 °C and then stained with a conjugated antibody against 1:80 cKIT (Anti-Hu CD117 (cKIT) (APC), Invitrogen #17-1179-42) and 1:80 P75 (anti-CD271 (FITC), BioLegend #345104). Cells double positive for FITC (P75) and APC (cKIT) were sorted and 4, 6-diamidino-2-phenylindole (DAPI) was used to exclude dead cells.

## Melanoblast expansion

At day 11, melanoblasts were aggregated into 3D spheroids (2 million cells/well) in ultra-low attachment 6-well culture plates (corning, 3471). Cells were expanded for maximum 7 days and then used for the Seahorse experiments.

Melanoblast media:
Neurobasal media (gibco, 21103-049)
1 mM L glutamine (gibco, 25030-081)
0.1 mM MEM NEAA (gibco, 11140-050)
FGF2 10 ng/ml (R&D, 233-FB/CF)
CHIR 3uM (R&D, 4423)
B27 supplement (gibco, 12587-010)
N2 supplement (gibco, 17502-048)

## Melanocyte differentiation

Upon FACS sorting, P75$^+$cKIT$^+$ melanoblasts were plated onto dried PO/Lam/FN dishes. Cells were fed with melanocyte medium every 2–3 days. Cells were passaged once a week at a ratio of 1:6, using accutase for 20 min at 37 °C for cell detachment. Mature melanocytes at day 100 were used for the seahorse experiments.

Melanocyte media (-1 L):
Neurobasal media 500 ml (gibco, 21103-049)
DMEM/F12 500 ml (gibco, 11330-032)
SCF 25 ng/ml (R&D, 255-SC-MTO)
cAMP 250uM (Sigma, D0627)
FGF2 5 ng/ml (R&D, 233-FB/CF)
CHIR 1.5uM (R&D, 4423)
BMP4 12.5 ng/ml (R&D, 314-BP)
EDN3 50 nM (Bachem, 4095915.1000)
25 ml FBS (R&D, S11150H)
2.5 ml penicillin/streptomycin (gibco, 15140-122)
2 ml L-Glutamine (gibco, 25030-081)
B27 supplement (gibco, 12587-010)
N2 supplement (gibco, 17502-048)

## Induction of melanocytic cell state in A375 cell line

Human A375 cells were trypsinized, centrifuged at 300 g for 3 min, and counted for viability then 500,000 viable cells per well were seeded in a six well TC treated plate. After 6 h and confirming cell attachment, media was aspirated, and fresh media was added with either DMSO, 200 µM IBMX (Cayman Chemical, 13347), or 20 µM Forskolin (EMD Millipore, 344270). Cells were incubated in drugs for 72 h and then harvested for qRT-PCR or Seahorse Mito Stress Test.

## qRT-PCR

Total RNA was isolated using the Quick-RNA Miniprep Kit (Zymo, R1055) according to the manufacturer's instructions. cDNA was synthesized using SuperScript IV First-Strand Synthesis System (Thermo Fisher, 18091200) and qPCR was performed using Applied Biosystems PowerUp (Thermo Fisher, A25742). Results were normalized to the *beta-actin* housekeeping gene.

## Human primer Sequences

*dct* forward - TCGATCTGCCAGTTTCAGTT
*dct* reverse - GAGCACCCTAGGCTTCTTCT
*pmel* forward - CAGTGTCTGGGCTGAGCATT
*pmel* reverse - GAGAAAGGCACCTGGTCAGT
*beta-actin* forward - CACCAACTGGGACGACAT
*beta-actin* reverse - ACAGCCTGGATAGCAACG

## Seahorse Mito Stress Test

The Seahorse XF Mito Stress Test (Agilent, 103015) was performed using the Seahorse XFe96 Analyzer. For melanoblasts and melanocytes, cells 30,000 melanoblasts and 10,000 melanocytes were seeded per well in an XF cell plate as previously described[13]. For human A375 cells, cells were treated with DMSO, IBMX, or Forskolin as described above then cells were trypsinized and resuspended in drug-containing media at 30,000 cells per well in an XF cell plate coated with 0.05% poly-L-lysine then incubated overnight (Sigma-Aldrich, 4707).

Cells were incubated in XF Mito Stress Test assay medium (Seahorse XF DMEM medium, pH 7.4, 10 mM glucose, 2 mM glutamine, 1 mM sodium pyruvate) for 1 h prior to measurement in a $CO_2$ free incubator at 37 °C. During assay run, cells were exposed to 2.0 µM oligomycin, 2.0 µM FCCP, and 0.5 µM rotenone/antimycin A. OCR and ECAR were normalized to nuclei fluorescence unit (FU) via SYTO 24 (Thermo Fisher, S7559) or protein via Pierce BCA Protein Assay Kit (Thermo Fisher, 23227) as indicated. Experimental measurements were analyzed using the Agilent Wave software.

## Fatty acid uptake

RPMI-7951, A2058, SKMEL5, and SKMEL28 cells were seeded at 8000 cells per well in a 96-well TC-treated black microplate (Greiner Bio-One, 655090). Cells were incubated with DMEM 10% or 1% FBS for 24 h. Cells were washed three times with serum-free DMEM and incubated in serum-free DMEM for 2 h.

A375 cells were seeded at 30,000 cells per well in a 96 well TC treated black microplate and incubated with DMSO, 200 µM Forskolin, or 20 µM IBMX for 24 h. Cells were washed three times with DMEM 1% FBS and incubated in DMEM 1% FBS with DMSO, Forskolin, or IBMX for one hour.

Lipid uptake was measured using the QBT Fatty Acid Uptake Assay (VWR, 10048-826) using the BioTek Synergy plate reader. For RPMI-7951, A2058, SKMEL5, and SKMEL28 cells, fluorescence at 485 nm excitation and 528 nm emission was measured every 50 s, for 600 s[35]. Relative fluorescence unit (RFU) was measured and the slope of the line for RFU per second was quantified across cell lines to determine uptake velocity. For A375 cells, fluorescence was measured every 50 s for 30 min. Fold change fatty acid uptake was measured by normalizing fluorescence at each time point to the fluorescence at the start of the assay to determine relative uptake over time[35].

## Seahorse Fatty Acid Oxidation Test

A375 cells were trypsinized, centrifuged at 300 g for 3 min, and counted for viability then 500,000 viable cells per well were seeded in a six well TC-treated plate. After 6 h and confirming cell attachment, media was aspirated and fresh media was added with either DMSO, 200 µM IBMX, or 20 µM Forskolin. Cells were incubated in drugs for 48 h and then tested for fatty acid oxidation in a protocol adapted from the Seahorse XF Palmitate Oxidation Stress Test. Cells were trypsinized and resuspended in DMEM 10% FBS at 30,000 cells per well in an XF cell plate coated with 0.05% poly-L-lysine (Sigma-Aldrich, 4707). Cells were allowed to adhere to the plate for 5 h then washed twice with Seahorse XF DMEM medium, pH 7.4. Cells were incubated overnight in DMSO, 200 µM IBMX, or 20 µM Forskolin in nutrient-

limited media: Seahorse XF DMEM medium, pH 7.4, 5 mM glucose, 1 mM glutamine, 1% FBS, and 0.5 mM carnitine (Fisher Scientific, AC230280050). In addition, cells were supplemented with either 150 μM BSA (Sigma-Aldrich, A1595) or 150 μM oleic acid conjugated to BSA (Sigma-Aldrich, O3008).

We used the Seahorse XF Long Chain Fatty Acid Oxidation Stress Test kit (Agilent, 102720-100). Cells were incubated in assay media supplemented with 150 μM BSA or oleic acid conjugated to BSA as fatty acid substrate in the following formulation: Seahorse XF DMEM medium, pH 7.4, 5 mM glucose, 1 mM glutamine, 0.5 mM carnitine. Cells were incubated for 1 h prior to measurement in a $CO_2$ free incubator at 37 °C. During assay run, cells were exposed to 10 μM etomoxir, 1.5 μM oligomycin, 2.0 μM FCCP, and 0.5 μM rotenone/antimycin A. OCR and ECAR were normalized to nuclei FU via SYTO 24 (Thermo Fisher, S7559). Experimental measurements were analyzed using the Agilent Wave software.

## Triacylglycerol quantification
RPMI-7951, A2058, SKMEL5, and SKMEL28 cells were seeded at 8000 cells per well in a 96 well TC treated black microplate for 24 h. Cells were incubated with DMEM 1% FBS supplemented with either 100 μM BSA or oleic acid conjugated to BSA for 24 h. Triglyceride levels were quantified using the Triglyceride-Glo Assay (Promega, J3160) as per manufacturer instructions.

## Proliferation assay on cell lines
RPMI-7951, A2058, SKMEL5, and SKMEL28 cells were seeded at 2500 cells per well in a 96 well TC treated black microplate for 24 h. Every 24 h for 72 h total, cells were incubated with fresh DMSO or DGAT1 inhibitor (T863, Cayman Chemical Company, 25807) in nutrient-limited media (DMEM supplemented with 5 mM glucose, 1 mM glutamine, 1 mM sodium pyruvate, and 1% FBS). Cell proliferation was quantified using CyQUANT Direct Red Cell Proliferation Assay (Thermo Fisher, C35013) per manufacturer's instructions.

## Seahorse ATP rate assay
The Seahorse XF Real-Time ATP Rate Assay (Agilent, 103592) was performed using the Seahorse XFe96 Analyzer. Cells were seeded at 5000 cells per well in an XF cell plate coated with 0.05% poly-L-lysine and then incubated overnight. Every 24 h for 72 h total, cells were incubated with fresh DMSO or 40 μM DGAT1 inhibitor in nutrient-limited media (DMEM supplemented with 5 mM glucose, 1 mM glutamine, 1 mM sodium pyruvate, and 1% FBS). Cells were incubated in assay medium (Seahorse XF DMEM medium, pH 7.4, 10 mM glucose, 2 mM glutamine, 1 mM sodium pyruvate) for 1 h prior to measurement in a $CO_2$-free incubator at 37 °C. During assay run, cells were exposed to 1.5 μM oligomycin and 0.5 μM rotenone/antimycin A. OCR, ECAR, and Proton Efflux Rate were normalized to nuclei FU via SYTO 24 (Thermo Fisher, S7559). Experimental measurements were analyzed using the Agilent Wave software generating the mitoATP and glycoATP measurements.

## Lipid droplet staining and imaging
Cells were seeded in each well of the Millicell EZ slide four-well (EMD Millipore, PEZGS0416). A2058 and SKMEL5 cells were seeded at 50,000 cells per well while RPMI-7951 and SKMEL28 cells were seeded at 30,000 cells per well. After 24 h, media was changed with DMEM 1% FBS supplemented with 10 μM BODIPY FL C16 (Thermo Fisher, D3821) and 10 μM BSA. 24 h after lipid uptake, cells were fixed with 4% formaldehyde for 15 min. For human A375 cells, 40,000 cells were seeded in each well with regular media and DMSO, IBMX, or Forskolin. After 72 h, cells were fixed with 4% formaldehyde for 15 min, washed with 1x PBS, and permeabilized in 0.1% Triton-X 100 in PBS for 30 min. Cells were washed, blocked with 10% goat serum (Thermo Fisher, 50-062Z), and incubated overnight at 4 °C in 1:100 dilution of rabbit anti-PLIN2

(Proteintech, 15294-1-AP). Cells incubated for 1 h at room temperature in 1:500 goat anti-rabbit Alexa Fluor 555 (Cell Signaling Technology, 44135), 1:400 Alexa Fluor 488 Phalloidin (Thermo Fisher, A12379) and 1:2000 Hoechst 33342 (Thermo Fisher, H3570).

For zebrafish ZMEL-LD cells, 250,000 cells were seeded in each well of the Millicell EZ slide four-well and incubated with 100 μM of oleic acid for 24 h. For cells treated with DGAT1 inhibitor, cells were treated with 20 μM T-863 (Cayman Chemicals, 25807). Cells were fixed with 4% formaldehyde for 15 min, washed with 1x PBS, and stained with 1:2000 Hoescht for 10 min.

Slides were mounted in Vectashield Antifade Mounting Media (Vector Laboratories, H-1000). Cells were imaged on the Zeiss LSM 880 inverted confocal microscope with AiryScan using a 63× oil immersion objective. Confocal stacks were visualized in FIJI 2.1.0. A375 melanoma lipid droplets were counted using FIJI. Lipid droplet per area was quantified in MATLAB for RPMI-7951, A2058, SKMEL5, and SKMEL28.

## CRISPR/Cas9 knockout in ZMEL-LD
The Alt-R CRISPR/Cas9 System (Integrated DNA Technologies) and Neon Transfection System were used according to manufacturer protocols for CRISPR/Cas9 knockout in ZMEL-LD cells. Knockout was validated by harvesting genomic DNA with the DNeasy Blood & Tissue Kit (Qiagen, 69506), *dgat1a* and *dgat1b* genomic loci were PCR amplified with Platinum SuperFi II PCR Master Mix (Thermo Fisher, 12368010) and indels detected using the Surveyor Mutation Kit (Integrated DNA Technologies, 706020). Cells were incubated overnight with 100 μM oleic acid and then analyzed for lipid droplets via flow cytometry as previously described[41]. Briefly, cells were stained for viability with 1:1000 DAPI, data acquired using the Beckman Coulter CytoFLEX Flow Cytometer (Beckman Coulter) and analyzed using FlowJo 10.8.1 (BD Biosciences).

## sgRNA target sequences
sgNT - AACCTACGGGCTACGATACG
sgDGAT1 #1 (dgat1a) - GTGACTCAAGCCAAACGCGG
sgDGAT1 #1 (dgat1b) - CAAAAGCGGACACAAGGCGC
sgDGAT1 #2 (dgat1a) - AGGGCTCGGCGAAGCACCGG
sgDGAT1 #2 (dgat1b) - CTCACCTCATTCTGTCGTAG

## Surveyor Nuclease Assay Primer - Primer sequence
*dgat1a* forward - CATTGGCTGTACCTGAATGTGT
*dgat1a* reverse - AGAAACGAGAAGGGCTCGG
*dgat1b* forward #1 - CTGCCTGGACTCGGTTTATTTA
*dgat1b* reverse #1 - GGTCGCATTTTCTCTTGTTTTC
*dgat1b* forward #2 - GAAAAACTTGCAGCTCAACGA
*dgat1b* reverse #2 - CTGTGCCATAGGCTACTGTACG

## CrispRVariants knockout
Zebrafish tumors were dissected and sorted for tdTomato+ cells as described above then harvested for genomic DNA with the DNeasy Blood & Tissue Kit and the *dgat1a* locus was PCR amplified using primers from surveyor nuclease assay. DNA was purified using the NucleoSpin Gel & PCR Clean-up Kit (Takara, 740986.20) and 100 bp paired-end reads were sequenced using Illumina NovaSeq at the Integrated Genomics Operation at MSKCC. Sequences were aligned and percent of indels at the target site was quantified using R version 4.0.5 and CrispRVariants 1.18.0 as previously described[47].

## Statistics and reproducibility
Statistical analysis and figures were generated in GraphPad Prism 9.1.1, R Studio 4.0.5, and Biorender.com. Image processing and analysis were performed in MATLAB R2020a and FIJI 2.1.0. Statistical tests and p-values are reported in the figure legend for each experiment. Experiments were performed at least three independent times unless

noted in the figure legend for each experiment. Sample sizes were chosen based on previous studies using zebrafish transgenesis and limited by the number of embryos produced on any given day.

## Reporting summary
Further information on research design is available in the Nature Portfolio Reporting Summary linked to this article.

## Data availability
The sequencing data generated in this study have been deposited in the Gene Expression Omnibus database under accession code GSE201378. The processed sequencing data are provided in Supplementary Tables. The sequencing data and key findings generated in this study are provided in the Supplementary Information/Source Data file. The single-cell RNA-sequencing data from Smalley et al. used in this study[24] are available in the GEO database under accession code GSE174401. The CCLE and TCGA data used in this study are available in the Depmap database [https://depmap.org/portal/download/all/] and TCGA database [https://tcga.xenahubs.net]. Source data are provided as a Source Data file. Source data are provided in this paper.

## Code availability
Code used to analyze data from this paper has been uploaded to https://github.com/dlumaquin/LDmelanoma.

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

## Acknowledgements

We thank the Aquatics Services and Systems facility, A. Afolalu, and M. Shepard for zebrafish maintenance. The Integrated Genomics Organization and Single Cell Research Initiative with R. Chaligne, O. Chaudhary, and N. Sohail provided technical support for bulk and single-cell RNA sequencing. The Flow Cytometry Core Facility provided technical support for FACS experiments. The Molecular Cytology Core Facility (P30CA008748) provided confocal imaging support. Donald B. and Catherine C. Marron Cancer Metabolism Center, J. Cross, and S.J. Raman provided technical and conceptual support for Seahorse metabolic assays.

R.M.W. was funded through the NIH/NCI Cancer Center Support Grant P30 CA008748, the Melanoma Research Alliance, The Debra and Leon Black Family Foundation, NIH Research Program Grants R01CA229215 and R01CA238317, NIH Director's New Innovator Award DP2CA186572, The Pershing Square Sohn Foundation, The Mark Foundation for Cancer Research, The American Cancer Society, The Alan and Sandra Gerry Metastasis Research Initiative at the Memorial Sloan Kettering Cancer Center, The Harry J. Lloyd Foundation, Consano and the Starr Cancer Consortium (all to R.M.W.). D.L. was supported by NIH Kirschstein-NRSA predoctoral fellowship (F30CA254152). D.L., C.L., and Y.M were supported by NIH Medical Scientist Training Program grant (T32GM007739-42). E.M. was supported by NIH Individual Predoctoral to Postdoctoral Fellow Transition Award (5K00CA223016-04). A.B. was supported by the GMTEC Scholars Gerry Fellowship. Y.M. was supported by NIH Kirschstein-NRSA predoctoral fellowship (F30CA265124) and Barbara and Stephen Friedman Pre-doctoral Fellowship. S.S. was supported by a Melanoma Research Foundation Career Development Award (719502) and MSKCC TROT Grant (T32CA160001).

## Author contributions

D.L. and R.M.W. developed the experiments and interpreted the results. D.L. performed most experiments and collected and analyzed data. D.L., E.M., Y.M., and S.S. performed zebrafish-related experiments. Y.M. assisted with computational analyses. E.J., C.L., and T.H. assisted with human cell line-related experiments. A.B. and L.S. performed and provided hPSC experiments and reagents. D.L. and R.M.W. wrote the manuscript. R.M.W. acquired funding for the project. All authors read and edited the manuscript.

## Competing interests

R.M.W. is a paid consultant to N-of-One Therapeutics, a subsidiary of Qiagen. R.M.W. receives royalty payments for the use of the casper line from Carolina Biologicals. L.S. is co-founder and consultant of BlueRockTherapeutics and is listed as an inventor on a patent application by Memorial Sloan Kettering Cancer Center related to melanocyte differentiation from human pluripotent stem cells (WO2011149762A2). D.L., E.M., E.J., A.B., Y.M., C.L., T.H., and S.S. declare no competing interests.
