## [Peer Review File · Nature Communications]

Lipid droplets are a metabolic vulnerability in melanomaREVIEWER COMMENTS

Reviewer #1 (Remarks to the Author):

Lumaquin et al made use of a BRAFV600/p53-/-/PTENko zebrafish melanoma model and single cell sequencing to describe different melanoma phenotypes. They identified a melanocytic phenotype characterized by increased oxidative phosphorylation and lipid droplet formation. The authors' main claim is that the melanocytic cell state is dependent on lipid droplets. Accordingly, they show that the oxidative metabolism is reduced upon suppression of lipid production. To translate their findings to in vivo, the authors perturbed lipid droplet formation by targeting DGAT1 using CRISPR technology. This resulted in (limited) tumor growth suppression in their melanoma zebrafish model.

Major points:

1. Limited cell line systems to support quite major claims.

1a. Although they start out in their well-known zebrafish model, the authors base many of their in vitro findings on a single human cell line model, A375 (Figure 2C-E, 3C and 5D). They could (should) have used a panel of cell lines that are phenotypically different to strengthen their claims. They could also mine databases to look for supportive data. For example, Tsoi et al. (2018) derived their different melanoma signatures from a panel of cell lines. Furthermore, the cancer cell line encyclopedia offers data (RNAseq, metabolomics, proteomics and more) in a large set of cell lines. Additionally, the authors could put together their own panel.

1b. The authors use a human cell line, A375, which does not present a melanocytic phenotype intrinsically. Instead, this had to be induced by IBMX or forskolin treatment (Figure 2C-E, 3C and 5D). Although a model like this may be used as supportive data, I wonder why the authors chose to not use their zebrafish system to study this question. Furthermore, the presence of the drugs could have an influence on the readout independently of the phenotypic change, which is not controlled for.

1c. The upregulation of dct and pmel is not sufficient evidence that A375 is melanocytic (Figure 2D). To interpret these results the authors should add at least an additional cell line that is melanocytic to compare pmel & dct levels. In addition, determining the degree of melanin production may be informative.

1d. The authors use a seemingly confounding human pluripotent stem cell model (Figure 2A). They mention that seahorse experiments were performed with melanoblasts that were max. 7 days old.

However, this did not seem to be the case for melanocytes derived from the stem cells, which were at least 100 days old. The difference in age of the cells could contribute to the differences in OCR described by the authors, for example if senescence/aging would occur.

2: Novelty of the connection between the melanocytic cell state and oxidative phosphorylation. I am not entirely convinced that the association between the melanocytic cell state with oxidative phosphorylation is novel (Figure 1 and 2), as the authors acknowledge themselves in their Discussion. Similar findings have been described by Fischer, Haq, Gopal. For example, MITF, a key transcription factor of the melanocytic cell state, drives PGC1a-mediated oxidative phosphorylation. Could the authors better explain what exactly is the new piece of data that advances our insight over those previous studies?

3: Questions about data bias. The authors performed RNA-seq on zebrafish tumors with sgNT or sgDGAT1a (Figure 5). This approach may have yielded interesting results, given the big clouds observed in the volcano plot. However, the biggest changes were not shown, let alone followed up. It looks like the authors instead (subjectively) chose to highlight several (marginally) affected genes in the data. This suggests data bias, which I am sure was not the authors' intention, but which should be addressed thoroughly. I recommend the authors uncover the biggest changes observed and take a more unbiased approach to better support their claims.

4: Level of support for the strongest claim. The authors repeatedly focus on the potential of DGAT1/lipid droplets as a therapeutic target, specifically in melanoma. This observation is highlighted in the title and is repeated throughout their discussion. Dependency on DGAT1/lipid droplets in melanoma specifically is an interesting observation. However, their actual data supporting this is limited to panel 4c, where the authors show reduced fitness of sgDGAT1a melanoma in their zebrafish model. This effect was unfortunately quite limited and not pursued or validated. This reduces the enthusiasm of this referee for the outcome of this study. Furthermore, the authors did not show lipid droplets by immunohistochemistry in zebrafish tumors for both the control and DGAT1a knock-out condition (Figure 4C and D).

Minor points:

1. The order of the phenotype labels is different in Figure 1E compared to 1B and 1C.

2. In Figure 1B, 1C and 1E the authors present novel signatures, though they use the Tsoi signature for the melanocytic cell state. For consistency, it would be helpful to see whether the other 3 Tsoi signatures are also present in the single cell data.

3. It is unclear why it is relevant to look at single cell data of metastatic brain melanoma. The authors should explain the choice for the Smalley dataset (Figure S3A).
4. It is unclear why p53 was not deleted in the human pluripotent stem cell model (Figure 2a) for consistency with the zebrafish model (Figure 1a).
5. The fold change is small in Figure 2E (middle panel). One wonders whether such differences are biologically meaningful.
6. The normalization is different among Seahorse experiments presented (Figure 2B, 2E and 3a). This may affect or even confound the interpretation of these results.
7. The left and right plots do not match (Figure 3A). The authors placed additional data in the supplement.
8. Starting points of the different conditions used in the Fatty acid uptake assay are different (Figure 3B). Also, one wonders whether 24 hours incubation of A375 with IBMX or forskolin can induce the melanocytic cell state. The authors showed in Figure 2 an incubation time of 3 days in order to reach this cell state. At the least, this seems inconsistent.
9. The phenotype of the zebrafish is different between non targeting control and DGAT1a knockout condition. Furthermore, it seems that the intestine is more enlarged in the control condition (Figure 4D).
10. The authors do not show that ZMEL-LD has a melanocytic phenotype (Figure 4A).
11. The CRISPR knock-out does not cause RNA loss per se. Even if this is common, it is expected to be different for each guide (Figure 5B and in text comment on expectation).

Reviewer #2 (Remarks to the Author):

Lumaquin et al. use their own zebrafish melanoma model to identify 5 transcriptionally distinct melanoma cell “states,” one of which corresponded to pigmented melanocytic cells that were enriched

in expression of genes involved in oxidative metabolism. These cells are relevant to human melanoma in that they have been implicated in tumor progression due to their highly proliferative phenotype. The authors use a combination of hPSC-derived melanoblasts and melanocytes, a well-known human melanoma cell line (A375), and their Zebrafish electroporation model together with single-cell sequencing, RNA-seq, gene expression analyses, genetic engineering, histology, and fatty acid oxidation assays (Seahorse) to develop a model of melanoma tumor progression that is dependent on mitochondrial fatty acid oxidation in pigmented melanocytes and reveals a metabolic vulnerability that might be targetable therapeutically. This is a potentially high impact contribution but with some moderation of enthusiasm based on the comments below.

1- There appears to be a much bigger difference in oxidative phenotype between hPSC-derived melanoblasts and melanocytes versus that between vehicle and IBMX or forskolin treated A375 cells. Thus, it is unclear how representative the IBMX/forskolin-treated A375 melanoma line is of pigmented melanocytes. Do they represent a true altered transcriptional “state” or simply a cAMP-driven transcriptional effect. The pigmented melanocytes identified by single-cell sequencing existed in a milieu of other melanoma cell transcriptional states, all of which were in the same extracellular environment (i.e. no added IBMX or forskolin). This issue reduces the impact of the subsequent functional studies in A375 cells in Fig. 3.

2- In Fig. 3c, puncta marked by the Proteintech PLIN2 antibody are assumed to be lipid droplets, which is likely the case. The case would be strengthened by using a stain for neutral lipid (oil Red O or Bodipy), as well. Increased numbers of the presumed lipid droplets are shown to occur with IBMX, forskolin and oleic acid treatment, but do these treatments lead to increased triglyceride content, new lipid droplets (oleate) or dispersion of pre-existing droplets (IBMX, forskolin)?

3- Fig. 4 addresses the question of whether preventing lipid droplet formation by knockout or inhibition of DGAT1 affects tumor progression. A highly complementary approach would be knockdown or inhibition of ATGL. In such case, the lipid droplets would be expected to enlarge but fatty acids would not be released from the droplets. The results would permit the refinement of the conclusion to include differentiation between effects of the lipid droplets versus effects of the fatty acids. Lipid droplets have been shown in other systems to sequester histones, so it is conceptually possible these organelles could regulate gene expression indirectly.

4- Are MITF and PGC1alpha increased in the pigmented melanocytic state versus the other less oxidative states? MITF is implicated in the Discussion, so any data in support or against this potential mechanism should be provided.

5- It is surprising that the authors do not investigate or discuss PLIN5 in addition to PLIN2, because PLIN5 is expressed most highly in oxidative tissues. Is PLIN5 induced in the melanocytic hPSCs compared with the melanoblastic hPSCs? PLIN5 can promote the PGC1alpha gene program in the nucleus to drive mitochondrial beta oxidation. There are 4 PLINs expressed in zebrafish, plin1, plin2, plin345, and plin6. plin6 has been identified in zebrafish and it is highly expressed in skin xanthophores (Granneman et al. eLife 2017).

6- Do the authors think that the role of oxidative metabolism in melanoma progression applies to all melanomas or to just those with the BRAFV600E mutation?

Reviewer #3 (Remarks to the Author):

This intriguing paper by Lumaquin et al. analyzes metabolic and phenotypic heterogeneity in melanoma tumors. There are several aspects of this study that are novel and of great interest to cancer metabolism researchers and the zebrafish community. Using TEAZ, the authors create tumors that express BRAFV600E and are PTEN deficient in p53KO immunocompetent adult zebrafish. This tumor model recapitulates genetic changes that are common in melanoma, but with expression restricted to adult animals. Additionally, the authors employ a set of technologies, including scRNA-seq and human iPSCs, to investigate the metabolic mechanisms underlying phenotypic heterogeneity in a population of tumor cells that have a primarily melanocytic phenotype. I found this to be very interesting and exciting.

The oxidative phenotype for cells with a melanocytic cell state is very well supported by the transcriptional profiling and the Seahorse studies. Enrichment of both oxidative phosphorylation and fatty acid metabolism is seen in the analysis. These pathways can encompass many different processes. Can the authors comment on the genes enriched in the fatty acid metabolism module: is CPT1 or other mitochondrial FA genes in that module or is it primarily lipid biosynthetic genes? This may lend further support to their hypothesis that oxidative phosphorylation is driven by the degradation of lipids.

Similarly, in Fig 3 and supplementary Fig 5, their A375 cells have a modest but significant impact on OCR after CPT1 inhibition with etomoxir treatment. The result suggests that glucose (or other fuels such as glutamine, especially considering there is a lot of glutamine present in the media) are also being used to sustain oxidative phosphorylation. The scRNA-seq might provide additional information on what fuels are used. It would be interesting if the authors could add a comment or two about this into the paper.

The *in vivo* deletion of DGAT1 and subsequent reduction in tumor growth is a nice validation that lipid droplets contribute to the proliferation of melanocytic cells. The authors include compelling data after inhibition or knockout of DGAT1 in the ZMEL cell line. It would be nice if the authors could also examine the lipid droplets within tumors by histology to rule out that these cells can still make lipid droplets, especially given that *de novo* lipid biosynthesis appears to be upregulated.

Fig 5D uses Seahorse to examine the metabolic consequences of knockdown in DGAT1 in A375. The reduction in both ATP generated by glycolysis and oxidative phosphorylation is interesting. Typically, a reduction in mitochondrial ATP production would result in an increase in glycolysis with subsequent

production of lactate. Can the authors comment on whether this is the case or if elimination of lipid droplets leads to alterations in glycolysis as well? Perhaps the authors could speculate on a potential mechanism for this fascinating observation?

A small note, on line 122, "FAO can be fueled by either de novo synthesis or uptake from extracellular sources". Canonically FAO and FAS don't function simultaneously as it would be a futile cycle. But there is some evidence that cells might actually do this (PMID: 27049668). Also, cancer cells can take up exogenous FA for beta-oxidation or cancer cells may take up exogenous FA and import them into mitochondria to remodel the mitochondrial membrane. CPT1 may play a role in both processes.

I hope these comments and suggestions will be useful during revision. Overall, this is an impressive study and I believe that it is well suited for publication in Nature Communications.

Reviewer #4 (Remarks to the Author):

The authors showed a relationship between lipid droplet and melanocytic cell in melanoma from the metabolic points of view. The fact that this study focused on the role of pigmented, melanocytic cell state which requires further research makes it unique as it can be used for better understanding of clinical data. Using melanoma-generating techniques in zebrafish, in vivo studies were done and demonstrated that inhibiting lipid droplet generation induces reduction of late stage tumor. But there may still be some unanswered questions that the authors could help to address using more experiments or discussion.

Here are my comments:

Major

1. As the authors had mentioned, the correlation between lipid droplet and melanoma is already well known. Although explained from the metabolic points of view, it is not clear what is new finding.
2. As shown in figure 4, the degree of tumor size reduction doesn't seem significant enough when the *dgat1* gene was knocked out. It would be nicer if authors can perform additional experiments to show clear inhibition of cancer growth.

Minor

1. In figure 1f, what about the other pathways among the top 10 GO biological processes? The authors mainly focused on the oxidative phosphorylation / fatty acid metabolic process which are quite low in enrichment score rank. It seems a bit unnatural as top scoring pathways were excluded for further experimental focus. Please add some explanation about it.
2. In line 104~108, the experiment results are out of place and not fitting the figure 2C~2E, thus those sentences need rearrangement.
3. For figure 5, additional single-cell experiments could possibly increase the resolution of melanocytic cell state in melanoma although it's not necessary.

Reviewer #1 (Remarks to the Author):

Lumaquin et al made use of a BRAFV600/p53-/-/PTENko zebrafish melanoma model and single cell sequencing to describe different melanoma phenotypes. They identified a melanocytic phenotype characterized by increased oxidative phosphorylation and lipid droplet formation. The authors' main claim is that the melanocytic cell state is dependent on lipid droplets. Accordingly, they show that the oxidative metabolism is reduced upon suppression of lipid production. To translate their findings to in vivo, the authors perturbed lipid droplet formation by targeting DGAT1 using CRISPR technology. This resulted in (limited) tumor growth suppression in their melanoma zebrafish model.

Major points:

1. Limited cell line systems to support quite major claims.

1a. Although they start out in their well-known zebrafish model, the authors base many of their in vitro findings on a single human cell line model, A375 (Figure 2C-E, 3C and 5D). They could (should) have used a panel of cell lines that are phenotypically different to strengthen their claims. They could also mine databases to look for supportive data. For example, Tsoi et al. (2018) derived their different melanoma signatures from a panel of cell lines. Furthermore, the cancer cell line encyclopedia offers data (RNAseq, metabolomics, proteomics and more) in a large set of cell lines. Additionally, the authors could put together their own panel.

We have taken both a data mining approach as well as tested new representative cell lines. To identify cell lines and patient samples that better represented the entire spectrum of melanoma states, we utilized the data from the Celligner pipeline, which was optimized to integrate TCGA and CCLE datasets (Warren, *Nature Communications* 2021). This method allows for identification of gene signatures that are conserved across large scale human patient samples or cell lines. The integrated analysis (encompassing a total of n=38 cell lines and n=443 patient samples) revealed that melanomas fall into undifferentiated/neural crest, transitory and melanocytic states. In both the TCGA and CCLE datasets, the melanocytic cell state was associated with a significant enrichment for the fatty acid oxidation score (Figure 3a, Figure 3b and Supplementary Figure 5b,d).

From the CCLE data, we then selected an additional n=4 cell lines for further functional analysis as you suggested. We chose 2 cell lines classified as undifferentiated/neural crest (RPMI7951 and A2058) and 2 classified as melanocytic (SKMel5 and SKMel28). As described below in more detail, we found that the melanocytic cell lines had increased fatty acid uptake (Figure 3g), DGAT1 mRNA (Figure 4a), triglyceride levels (Figure 4b) and lipid droplet area (Figure 4c-f).

1b. The authors use a human cell line, A375, which does not present a melanocytic phenotype intrinsically. Instead, this had to be induced by IBMX or forskolin treatment (Figure 2C-E, 3C and 5D). Although a model like this may be used as supportive data, I wonder why the authors chose to not use their zebrafish system to study this question. Furthermore, the presence of the

drugs could have an influence on the readout independently of the phenotypic change, which is not controlled for.

As noted above, we have added an additional n=4 human cell lines that are intrinsically either undifferentiated or melanocytic, allowing us to study this without IBMX/forskolin induction. In addition, as noted below, we also used the zebrafish ZMEL1 melanoma line to test the effect of DGAT1 inhibition *in vivo*.

1c. The upregulation of *dct* and *pmel* is not sufficient evidence that A375 is melanocytic (Figure 2D). To interpret these results the authors should add at least an additional cell line that is melanocytic to compare *pmel* & *dct* levels. In addition, determining the degree of melanin production may be informative.

We agree. As noted above, the data from Celligner allowed us to more comprehensively analyze pigmentation genes in TCGA and CCLE datasets outside of A375 alone and functionally test the additional n=4 representative cell lines. Examination of *mita*, *dct*, *pmel* and *tyr* all show higher expression (Supplementary Figure 5) in the melanocytic lines (i.e. SKMe15, SKMe128, amongst numerous others) vs undifferentiated lines (i.e. RPMI7951, A2058 amongst others).

1d. The authors use a seemingly confounding human pluripotent stem cell model (Figure 2A). They mention that seahorse experiments were performed with melanoblasts that were max. 7 days old. However, this did not seem to be the case for melanocytes derived from the stem cells, which were at least 100 days old. The difference in age of the cells could contribute to the differences in OCR described by the authors, for example if senescence/aging would occur.

We have previously demonstrated the utility of human pluripotent stem cells for modeling melanocytic differentiation from the neural crest (Baggiolini, *Science* 2021). The major advantage of this approach is that the cells are isogenic to each other, which is not the case when comparing across cell lines (i.e CCLE) or patient samples (i.e. TCGA), where baseline germline and somatic genetic background will differ. However, we acknowledge that a limitation of the iPS system is that the melanoblasts are isolated at a much earlier time point than melanocytes. This is an inherent limitation of differentiation in all stem cell models, since more differentiated progeny will always take longer to be generated than progenitors. Despite this, this system provides an orthogonal way to look at progenitor vs. melanocytic cells that complements what can be done in established cancer cell lines. At 100 days, the melanocytic cells are still actively proliferating and do not show evidence of senescence, making it unlikely this alone explains the OCR phenotypes. Moreover, the association between the melanocytic state and oxidative metabolism we see with the iPS cells is in line with what we now observe in the 5 cancer cell lines we have studied. We have added a consideration of this limitation to the manuscript text.

2: Novelty of the connection between the melanocytic cell state and oxidative phosphorylation. I am not entirely convinced that the association between the melanocytic cell state with oxidative

phosphorylation is novel (Figure 1 and 2), as the authors acknowledge themselves in their Discussion. Similar findings have been described by Fischer, Haq, Gopal. For example, MITF, a key transcription factor of the melanocytic cell state, drives PGC1a-mediated oxidative phosphorylation. Could the authors better explain what exactly is the new piece of data that advances our insight over those previous studies?

As you point out, the previous work has linked MITF to PGC1a-mediated mitochondrial biogenesis to drive increased oxidative phosphorylation, but the substrates driving this and how the cell obtains those substrates is unknown. The goal of our study was to show that: 1) fatty acids are one key substrate, and that 2) this depends upon the uptake and processing in lipid droplets. While we cannot exclude that other substrates could be used, this data shows that fatty acids are especially important in the melanocytic state.

3: Questions about data bias. The authors performed RNA-seq on zebrafish tumors with sgNT or sgDGAT1a (Figure 5). This approach may have yielded interesting results, given the big clouds observed in the volcano plot. However, the biggest changes were not shown, let alone followed up. It looks like the authors instead (subjectively) chose to highlight several (marginally) affected genes in the data. This suggests data bias, which I am sure was not the authors' intention, but which should be addressed thoroughly. I recommend the authors uncover the biggest changes observed and take a more unbiased approach to better support their claims.

We agree this required a more clear and unbiased approach. In the new Figure 6, we show the top significantly enriched go biological pathways, which includes those both up and downregulated. The data shown in this Figure are the pathways ordered by adjusted p-value. As shown, downregulation of cell division and cell cycle genes are amongst the top pathways. We also provide a full analysis of all differentially expressed genes in Supplementary Data 2. It was this effect on cell cycle that led us to hypothesize that DGAT1 deficiency would be associated with decreased proliferation.

To further test this, we have now measured the effect of a DGAT1 inhibitor both in vivo and in vitro. We transplanted ZMEL1 melanoma cells into zebrafish and treated them with the DGAT1 inhibitor T863 (Cayman Chemical Company, 25807) or DMSO control. This revealed a significant decrease in overall tumor burden (Figure 6d). To test if this was directly due to proliferation, we then tested the DGAT1 inhibitor in vitro in both the undifferentiated (RPMI7951, A2058) and melanocytic (SKMel5 and SKMel28) cell lines described above and measured proliferation using CyQuant. While the DGAT1 inhibitor had minimal effect in the undifferentiated cells, we saw a significant decrease in proliferation in the melanocytic lines (Figure 6e). Together, this new data is consistent with the RNA-seq showing a decrease in proliferation with DGAT loss or inhibition. While we agree there are many other interesting pathways uncovered in our RNA-seq, these will need to be further characterized in the future.

4: Level of support for the strongest claim. The authors repeatedly focus on the potential of DGAT1/lipid droplets as a therapeutic target, specifically in melanoma. This observation is highlighted in the title and is repeated throughout their discussion. Dependency on DGAT1/lipid

droplets in melanoma specifically is an interesting observation. However, their actual data supporting this is limited to panel 4c, where the authors show reduced fitness of sgDGAT1a melanoma in their zebrafish model. This effect was unfortunately quite limited and not pursued or validated. This reduces the enthusiasm of this referee for the outcome of this study. Furthermore, the authors did not show lipid droplets by immunohistochemistry in zebrafish tumors for both the control and DGAT1a knock-out condition (Figure 4C and D).

We agree this needed stronger evidence. As mentioned above, we have now added in vitro and in vivo studies using a DGAT1 inhibitor. We find that the DGAT1 inhibitor inhibits proliferation in the melanocytic cell lines, and impairs tumor growth in zebrafish transplanted with the ZMEL1 melanoma cell line. We have also performed immunohistochemistry on the sgNT and sgDGAT1a tumors to probe for PLIN2. This is a protein found on the surface of the lipid droplet that we have previously shown specifically marks this organelle (Supplementary Figure 9).

Minor points:

1. The order of the phenotype labels is different in Figure 1E compared to 1B and 1C.

The order of clusters for Figure 1E is sorted in descending average melanocytic score. This is a conventional way of showing this for these analyses, although we can modify the order in categorical order according to b and c if the reviewer and editor prefer that.

2. In Figure 1B, 1C and 1E the authors present novel signatures, though they use the Tsoi signature for the melanocytic cell state. For consistency, it would be helpful to see whether the other 3 Tsoi signatures are also present in the single cell data.

Yes, we have found evidence of the other Tsoi signatures in our data, as shown below. As expected, the invasive state is most strongly associated with the undifferentiated or neural crest state.

3. It is unclear why it is relevant to look at single cell data of metastatic brain melanoma. The authors should explain the choice for the Smalley dataset (Figure S3A).

This was largely a practicality. The Smalley dataset is the largest single cell data set in melanoma, and allowed us to validate our results of increased oxidative metabolic signatures in more melanocytic populations. In addition, human melanomas that metastasize to the brain have been shown to upregulate oxidative metabolism (Fischer, *Cancer Discovery* 2019) and drugs targeting this are currently being explored to target these metastases.

4. It is unclear why p53 was not deleted in the human pluripotent stem cell model (Figure 2a) for consistency with the zebrafish model (Figure 1a).

We used the iPS system so that we could have isogenic cells at different states (i.e. melanoblast versus melanocyte) but did not focus on the role of p53 specifically here. We think it is unlikely that p53 explains the correlation between the melanocytic state and fatty acid oxidation, since it was not a consistent feature of either the n=5 cell lines we tested nor in the TCGA/CCLC Celligner analysis.

5. The fold change is small in Figure 2E (middle panel). One wonders whether such differences are biologically meaningful.

The magnitude of changes in Seahorse assays can often be subtle, since cells are typically not entirely oxidative or glycolytic. This is why the ratio of OCR/ECAR (reflecting oxidative vs glycolytic metabolism) are often more useful, since they show relative enrichment of one pathway over another.

6. The normalization is different among Seahorse experiments presented (Figure 2B, 2E and 3a). This may affect or even confound the interpretation of these results.

Depending on the configuration of each assay plate and treatment (i.e. # of cells, density), it was sometimes more practical to normalize using BCA normalization (i.e. protein) and in other cases to normalize using nuclear fluorescence normalization. For each experiment, the conditions are only compared against each other and not across experiments. For example, Figures 2b, 2e and 3a are not compared against each other. This is why we indicated which normalization method was used for each experiment on the y-axis.

7. The left and right plots do not match (Figure 3A). The authors placed additional data in the supplement.

We had initially separated the Seahorse traces for ease of visualization, but agree this made the Figure confusing. We kept the key bar graph in Figure 3 but moved all of the Seahorse traces into Supplemental Figure 6 for consistency.

8. Starting points of the different conditions used in the Fatty acid uptake assay are different (Figure 3B). Also, one wonders whether 24 hours incubation of A375 with IBMX or forskolin can induce the melanocytic cell state. The authors showed in Figure 2 an incubation time of 3 days in order to reach this cell state. At the least, this seems inconsistent.

The slight increase in starting points is likely due to the fact that the forskolin and IBMX conditions have more lipid droplets at baseline (Supplementary Figure 7), which would increase the rate of uptake of dye. This is further evidenced by higher uptake over time. In terms of timing, prior studies have shown that forskolin can activate MITF driven programs within 24 hours (i.e. Khaled, *Genes & Development* 2010), but we agree that this may not be enough time to fully induce pigmentation. It is for those reasons that we chose to test additional cell lines, as mentioned above, representing the undifferentiated vs. melanocytic state (i.e. no need for forskolin or IBMX).

9. The phenotype of the zebrafish is different between non targeting control and DGAT1a knockout condition. Furthermore, it seems that the intestine is more enlarged in the control condition (Figure 4D).

The original image was of a male and female fish, which is why they look so different. We have now replaced this with a new image that shows just the male fish for the sake of consistency (Figure 5b), but both male and female fish were included in the analysis.

10. The authors do not show that ZMEL-LD has a melanocytic phenotype (Figure 4A).

The ZMEL1 line has high levels of *mitfa*, and has been previously shown to rapidly pigment in both in vitro and in vivo conditions (Heilmann, *Cancer Research* 2015 and Kim, *Nature Communications* 2017).

11. The CRISPR knock-out does not cause RNA loss per se. Even if this is common, it is expected to be different for each guide (Figure 5B and in text comment on expectation).

We agree that this does not always happen with CRISPR. However, in this case, when we performed RNA-seq we found that there is a significant downregulation of *dgat1a* in the sgDGAT1a tumors. We have added the normalized counts in (Supplementary Figure 8).

Reviewer #2 (Remarks to the Author):

Lumaquin et al. use their own zebrafish melanoma model to identify 5 transcriptionally distinct melanoma cell “states,” one of which corresponded to pigmented melanocytic cells that were enriched in expression of genes involved in oxidative metabolism. These cells are relevant to human melanoma in that they have been implicated in tumor progression due to their highly proliferative phenotype. The authors use a combination of hPSC-derived melanoblasts and melanocytes, a well-known human melanoma cell line (A375), and their Zebrafish electroporation model together with single-cell sequencing, RNA-seq, gene expression analyses, genetic engineering, histology, and fatty acid oxidation assays (Seahorse) to develop a model of melanoma tumor progression that is dependent on mitochondrial fatty acid oxidation in pigmented melanocytes and reveals a metabolic vulnerability that might be targetable therapeutically. This is a potentially high impact contribution but with some moderation of enthusiasm based on the comments below.

1- There appears to be a much bigger difference in oxidative phenotype between hPSC-derived melanoblasts and melanocytes versus that between vehicle and IBMX or forskolin treated A375 cells. Thus, it is unclear how representative the IBMX/forskolin-treated A375 melanoma line is of pigmented melanocytes. Do they represent a true altered transcriptional “state” or simply a cAMP-driven transcriptional effect. The pigmented melanocytes identified by single-cell sequencing existed in a milieu of other melanoma cell transcriptional states, all of which were in the same extracellular environment (i.e. no added IBMX or forskolin). This issue reduces the impact of the subsequent functional studies in A375 cells in Figure 3.

We agree that the A375 cells alone were not an entirely robust model and the effects of IBMX and forskolin could be due to a cAMP-induced state. To better test this, we have added new experiments and analyses to more directly connect the melanocytic state to fatty acid oxidation. To identify cell lines and patient samples that better represented the entire spectrum of melanoma states, we utilized the data from the Celligner pipeline, which was optimized to integrate TCGA and CCLE datasets (Warren, *Nature Communications* 2021). This method allows for identification of gene signatures that are conserved across large scale human patient samples or cell lines. The integrated analysis (encompassing a total of n=38 cell lines and n=443 patient samples) revealed that melanomas fall into undifferentiated/neural crest, transitory and melanocytic states. In both the TCGA and CCLE datasets, the melanocytic cell state was associated with a significant enrichment for the fatty acid oxidation score (Figure 3a, Figure 3b and Supplementary Figure 5b,d).

From the CCLE data, we then selected an additional n=4 cell lines for further functional analysis. We chose 2 cell lines classified as undifferentiated/neural crest (RPMI7951 and A2058) and 2 classified as melanocytic (SKMel5 and SKMel28). As described below in more detail, we found that the melanocytic cell lines had increased fatty acid uptake (Figure 3g), DGAT1 mRNA (Figure 4a), triglyceride levels (Figure 4b) and lipid droplet area (Figure 4c-f).

2- In Figure 3c, puncta marked by the Proteintech PLIN2 antibody are assumed to be lipid droplets, which is likely the case. The case would be strengthened by using a stain for neutral lipid (oil Red O or Bodipy), as well. Increased numbers of the presumed lipid droplets are shown to occur with IBMX, forskolin and oleic acid treatment, but do these treatments lead to increased triglyceride content, new lipid droplets (oleate) or dispersion of pre-existing droplets (IBMX, forskolin)?

We have added additional data (Figure 4c-f) showing colocalization of PLIN2 with BODIPY. This is consistent with our prior publication showing that a PLIN2-tdTomato transgene extensively colocalizes with BODIPY signal (Lumaquin, Johns, *eLife* 2021).

Since, as you mentioned, the effects of forskolin/IBMX can be confounded by other effects, we studied triglyceride content using the new cell lines mentioned above. We found that oleic acid significantly increases triglyceride content, and this is enhanced in the more melanocytic lines compared to the undifferentiated lines (Figure 4b). Due to the small size of the lipid droplets in these cells, we do not currently have equipment that allows us to assess whether these triglycerides are shuttled into de novo lipid droplet biogenesis or transferred across existing lipid droplets.

3- Figure 4 addresses the question of whether preventing lipid droplet formation by knockout or inhibition of DGAT1 affects tumor progression. A highly complementary approach would be knockdown or inhibition of ATGL. In such case, the lipid droplets would be expected to enlarge but fatty acids would not be released from the droplets. The results would permit the refinement of the conclusion to include differentiation between effects of the lipid droplets versus effects of the fatty acids. Lipid droplets have been shown in other systems to sequester histones, so it is conceptually possible the these organelles could regulate gene expression indirectly.

We agree this is an interesting experiment. In previous work (Lumaquin, Johns, *eLife* 2021) we showed that the ATGL inhibitor Atglistatin could increase lipid droplet content in melanoma cell lines, suggesting this enzyme is active in this lineage. To test this *in vivo*, we therefore treated fish transplanted with ZMEL1 cells and measured tumor burden after Atglistatin treatment. Interestingly, we saw no effect on tumor burden in this setting. This suggests that, at least in this setting, it is the fatty acids that are playing a dominant role. However, we cannot exclude the possibility that the increase in lipid droplet content in this setting is not playing a more subtle role in tumor progression (i.e. by changing gene expression, as you suggested), which we agree would be an interesting area for future exploration.

4- Are MITF and PGC1alpha increased in the pigmented melanocytic state versus the other less oxidative states? MITF is implicated in the Discussion, so any data in support or against this potential mechanism should be provided.

As mentioned above, we used Celligner to identify both cell lines (CCLE) and patient samples (TCGA) that represent the undifferentiated/neural crest, transitory and melanocytic states. In both datasets, we find that there is a concomitant increase in both MITF and PGC1a of these transcripts across progression from undifferentiated to melanocytic states (Supplementary Figure 5).

5- It is surprising that the authors do not investigate or discuss PLIN5 in addition to PLIN2, because PLIN5 is expressed most highly in oxidative tissues. Is PLIN5 induced in the melanocytic hPSCs compared with the melanoblastic hPSCs? PLIN5 can promote the PGC1alpha gene program in the nucleus to drive mitochondrial beta oxidation. There are 4 PLINs expressed in zebrafish, plin1, plin2, plin345, and plin6. plin6 has been identified in zebrafish and it is highly expressed in skin xanthophores (Granneman et al. eLife 2017).

We mainly chose to study PLIN2 based on our prior work showing its robust expression in melanoma (Lumaquin, Johns, eLife 2021). We agree that PLIN5 could be playing an important role in these more oxidative melanoma cells. However, in our single cell RNA-seq (Figure 1), there were no mapped transcripts for PLIN3/4/5, suggesting these are likely expressed at very low levels. Similarly, there were no mapped transcripts for PLIN4/5 in our bulk RNAseq (Figure 6). There was a modest upregulation of PLIN6 but because this is mainly in xanthophores (and not melanocytes) we did not pursue this further.

6- Do the authors think that the role of oxidative metabolism in melanoma progression applies to all melanomas or to just those with the BRAFV600E mutation?

In our Celligner analysis, this included both BRAF and NRAS tumors, suggesting this is not unique to the BRAF state. Moreover, a recently published study also showed that DGAT1 could be oncogenic in the setting of NRAS (Wilcock, *Cell Reports* 2022).

Reviewer #3 (Remarks to the Author):

This intriguing paper by Lumaquin et al. analyzes metabolic and phenotypic heterogeneity in melanoma tumors. There are several aspects of this study that are novel and of great interest to cancer metabolism researchers and the zebrafish community. Using TEAZ, the authors create tumors that express BRAFV600E and are PTEN deficient in p53KO immunocompetent adult zebrafish. This tumor model recapitulates genetic changes that are common in melanoma, but with expression restricted to adult animals. Additionally, the authors employ a set of technologies, including scRNA-seq and human iPS cells, to investigate the metabolic mechanisms underlying phenotypic heterogeneity in a population of tumor cells that have a primarily melanocytic phenotype. I found this to be very interesting and exciting.

The oxidative phenotype for cells with a melanocytic cell state is very well supported by the transcriptional profiling and the Seahorse studies. Enrichment of both oxidative phosphorylation and fatty acid metabolism is seen in the analysis. These pathways can encompass many different processes. Can the authors comment on the genes enriched in the fatty acid metabolism module: is CPT1 or other mitochondrial FA genes in that module or is it primarily lipid biosynthetic genes? This may lend further support to their hypothesis that oxidative phosphorylation is driven by the degradation of lipids.

We assessed the leading edge genes in the fatty acid metabolism module. This includes genes such as ANXA1, PTGR1, GPX4, SLC27A2, GSTP1, ELOVL1, CAV1, CKB, LDHB, ENO1, IDH1, ACADM. These are more suggestive of fatty acid metabolism rather than synthesis. However, because single cell data can be sparse in terms of absolute counts, we also more carefully examined fatty acid oxidation using bulk RNA-seq from the CCLE and TCGA datasets. In both datasets, the melanocytic cell state was strongly enriched specifically for fatty acid oxidation (Figure 3a,b) including genes such as CPT2 and HADHA (although not CPT1a itself). We also noted an increase in PPARGC1A, the regulator of mitochondrial biogenesis. Thus these data are suggestive that the melanocytic state is more associated with fatty acid oxidation in mitochondria rather than de novo lipid biogenesis.

Similarly, in Fig 3 and supplementary fig 5, their A375 cells have a modest but significant impact on OCR after CPT1 inhibition with etomoxir treatment. The result suggests that glucose (or other fuels such as glutamine, especially considering there is a lot of glutamine present in the media) are also being used to sustain oxidative phosphorylation. The scRNA-seq might provide additional information on what fuels are used. It would be interesting if the authors could add a comment or two about this into the paper.

We agree, and feel this data argues that other substrates are able to sustain oxidative metabolism as the cells become more melanocytic. It seems likely that this is glutamine, since it is present in all of the different media preparations. We have added this point to the manuscript. We did try and glean further clues from the scRNA-seq but did not see a clear signature other than the fatty acid metabolism.

The in vivo deletion of DGAT1 and subsequent reduction in tumor growth is a nice validation that lipid droplets contribute to the proliferation of melanocytic cells. The authors include compelling data after inhibition or knockout of DGAT1 in the ZMEL cell line. It would be nice if the authors could also examine the lipid droplets within tumors by histology to rule out that these cells can still make lipid droplets, especially given that de novo lipid biosynthesis appears to be upregulated.

We have performed immunohistochemistry for PLIN2 from the TEAZ generated tumors, as we previously showed that this protein is a specific marker of lipid droplets (Lumaquin, Johns, *eLife* 2021). This data (Supplementary Figure 9) shows that while DGAT1 knockout reduces PLIN2 expressing lipid droplets, they are not entirely eliminated. This is expected since DGAT1 is only one part of the entire cascade of proteins required for LD formation.

Fig 5D uses seahorse to examine the metabolic consequences of knockdown in DGAT1 in A375. The reduction in both ATP generated by glycolysis and oxidative phosphorylation is interesting. Typically, a reduction in mitochondrial ATP production would result in an increase in glycolysis with subsequent production of lactate. Can the authors comment on whether this is the case or if elimination of lipid droplets leads to alterations in glycolysis as well? Perhaps the authors could speculate on a potential mechanism for this fascinating observation?

We are not entirely sure of the reasons for this. One possibility is that acute loss of DGAT1 leads to a global reduction in cellular metabolism. This is supported by our observations that DGAT1 knockout is associated with a downregulation of cell cycle genes (Figure 6b). We also have new data showing that treatment of melanocytic lines with DGAT1 inhibitors leads to a significant decrease in proliferation (Figure 6e) and decreased tumor burden (Figure 6c-d). To further explore this, we analyzed relative rates of glycolysis vs. oxidative metabolism in n=4 cell lines (RPMI7951, SKMel5, SKMel28, and A2058). In 3 of the 4 lines (Supplementary Figure 10), we find that acute pharmacologic inhibition of DGAT1 leads to a decrease in both mitoATP and glycoATP, consistent with the above hypothesis.

A small note, on line 122, "FAO can be fueled by either de novo synthesis or uptake from extracellular sources". Canonically FAO and FAS don't function simultaneously as it would be a futile cycle. But there is some evidence that cells might actually do this (PMID: 27049668). Also, cancer cells can take up exogenous FA for beta-oxidation or cancer cells may take up exogenous FA and import them into mitochondria to remodel the mitochondrial membrane. CPT1 may play a role in both processes.

This is an important point. What we meant to say (and this is now indicated in the manuscript text) is that cells can increase uptake or mobilize existing lipid stores for FAO (which in turn can come from previously de novo synthesized fatty acids or breakdown of other lipid species to fatty acids). In our system, the more melanocytic cells have increased rates of fatty acid uptake compared to the more undifferentiated cells (Figure 3e-g), suggesting this is the major route.

Interestingly, in our RNA-seq analysis of the TCGA/CCLE data (Supplementary Figure 5) we did not see a major upregulation of CPT1a in the melanocytic state although we did see an increase in CPT2. The potential differential role of these two enzymes would be an interesting area for future exploration.

I hope these comments and suggestions will be useful during revision. Overall, this is an impressive study and I believe that it is well suited for publication in Nature Communications.

Reviewer #4 (Remarks to the Author):

The authors showed a relationship between lipid droplet and melanocytic cell in melanoma from the metabolic points of view. The fact that this study focused on the role of pigmented, melanocytic cell state which requires further research makes it unique as it can be used for better understanding of clinical data. Using melanoma-generating techniques in zebrafish, in vivo studies were done and demonstrated that inhibiting lipid droplet generation induces reduction of late stage tumor. But there may still be some unanswered questions that the authors could help to address using more experiments or discussion.

Here are my comments:

Major

1. As the authors had mentioned, the correlation between lipid droplet and melanoma is already well known. Although explained from the metabolic points of view, it is not clear what is new finding.

You are correct that previous work has shown a dependency on fatty acids, although the role of the lipid droplet itself has remained elusive. Our work presents two new important findings: 1) That it is mainly the melanocytic cell state that is dependent upon fatty acid uptake and oxidation, and not just all melanoma cells in general. It is now well recognized that melanomas are composed of at least 5 distinct cell states (i.e. melanocytic, stressed, proliferative, invasive and inflammatory), and our data now highlight a distinct metabolic program in one of those states compared to the others, and 2) That the lipid droplet organelle (and not solely the fatty acids) are a dependency in melanoma.

2. As shown in figure 4, the degree of tumor size reduction doesn't seem significant enough when the *dgat1* gene was knocked out. It would be nicer if authors can perform additional experiments to show clear inhibition of cancer growth.

To further test this, we have now measured the effect of a DGAT1 inhibitor both in vivo and in vitro. We transplanted ZMEL1 melanoma cells into zebrafish and treated them with the DGAT1 inhibitor T863 (Cayman Chemical Company, 25807) or DMSO control. This revealed a significant decrease in overall tumor burden (Figure 6d). To test if this was directly due to proliferation, we then tested the DGAT1 inhibitor in vitro in both the undifferentiated (RPMI7951, A2058) and melanocytic (SKMel5 and SKMel28) cell lines described above and measured proliferation using CyQuant. While the DGAT1 inhibitor had minimal effect in the undifferentiated cells, we saw a significant decrease in proliferation in the melanocytic lines (Figure 6e). Together, this new data is consistent with the RNA-seq showing a decrease in proliferation with DGAT loss or inhibition.

Minor

1. In figure 1f, what about the other pathways among the top 10 GO biological processes? The authors mainly focused on the oxidative phosphorylation / fatty acid metabolic process which

are quite low in enrichment score rank. It seems a bit unnatural as top scoring pathways were excluded for further experimental focus. Please add some explanation about it.

The other top pathways were all related to pigmentation. For example, due to the way GO terms are created, even pathways such as “organic hydroxy compound metabolic process” mainly contains genes in the pigmentation pathway. Because most of these genes are well known, we did not think this was the most productive line of investigation, whereas the correlation of oxidative metabolism with the pigmented state was novel and unexplored.

2. In line 104~108, the experiment results are out of place and not fitting the figure 2C~2E, thus those sentences need rearrangement.

This has been corrected.

3. For figure 5, additional single-cell experiments could possibly increase the resolution of melanocytic cell state in melanoma although it's not necessary.

We agree this would have helped resolve this further, but think this would be a good experiment for future studies to more directly link DGAT1 loss to specific cell states.

REVIEWER COMMENTS

Reviewer #1 (Remarks to the Author):

Lumaquin et al made use of a BRAFV600/p53-/-/PTENko zebrafish melanoma model and single cell sequencing to describe different melanoma phenotypes. They identified a melanocytic phenotype characterized by increased oxidative phosphorylation and lipid droplet formation. The authors' main claim is that the melanocytic cell state is dependent on lipid droplets. Accordingly, they show that the oxidative metabolism is reduced upon suppression of lipid production. To translate their findings to in vivo, the authors perturbed lipid droplet formation by targeting DGAT1 using CRISPR technology. This resulted in (limited) tumor growth suppression in their melanoma zebrafish model.

Major points:

1. Limited cell line systems to support quite major claims.

1a. Although they start out in their well-known zebrafish model, the authors base many of their in vitro findings on a single human cell line model, A375 (Figure 2C-E, 3C and 5D). They could (should) have used a panel of cell lines that are phenotypically different to strengthen their claims. They could also mine databases to look for supportive data. For example, Tsoi et al. (2018) derived their different melanoma signatures from a panel of cell lines. Furthermore, the cancer cell line encyclopedia offers data (RNAseq, metabolomics, proteomics and more) in a large set of cell lines. Additionally, the authors could put together their own panel.

We have taken both a data mining approach as well as tested new representative cell lines. To identify cell lines and patient samples that better represented the entire spectrum of melanoma states, we utilized the data from the Celligner pipeline, which was optimized to integrate TCGA and CCLE datasets (Warren, Nature Communications 2021). This method allows for identification of gene signatures that are conserved across large scale human patient samples or cell lines. The integrated analysis (encompassing a total of n=38 cell lines and n=443 patient samples) revealed that melanomas fall into undifferentiated/neural crest, transitory and melanocytic states. In both the TCGA and CCLE datasets, the melanocytic cell state was associated with a significant enrichment for the fatty acid oxidation score (Figure 3a, Figure 3b and Supplementary Figure 5b,d).

From the CCLE data, we then selected an additional n=4 cell lines for further functional analysis as you suggested. We chose 2 cell lines classified as undifferentiated/neural crest (RPMI7951 and A2058) and 2 classified as melanocytic (SKMel5 and SKMel28). As described below in more detail, we found that the melanocytic cell lines had increased fatty acid uptake (Figure 3g), DGAT1 mRNA (Figure 4a), triglyceride levels (Figure 4b) and lipid droplet area (Figure 4c-f).

Reviewer 1 comments on rebuttal:

I appreciate that the authors made the effort to provide considerable additional data supporting their in vitro findings. First of all, they added fatty acid oxidation gene scores from the CCLE and TCGA database which further supports their claims. Furthermore, the authors included several experiments in a panel of cell lines with different phenotypes. Since such significant changes have been made to the figures and panels 3, 4 and s5 I have carefully evaluated the new data and listed my comments below.

1. Figure 3a, b: Is there overlap between the fatty acid oxidation gene set and the melanocytic gene set? If so, this could confound the correlation.

2. Figure 3e: The IBMX and Forskolin conditions already have an increase at time point zero, even though the authors note the data was normalized to time point zero.

3. Figure 3f:

a. It would strengthen the data if the authors would include neural crest and undifferentiated markers.

b. A2058 is not a compelling neural crest like cell line and I feel should be characterized as melanocytic transitory.

4. Figure 3g and figure s5: The authors start the y-axis at arbitrary numbers without clearly indicating this with the correct symbol (broken axis with two dashed lines). This should be corrected because it could lead to incorrect interpretation of the data.

5. Figure 3g & 4b: The validation of figure 3e in figure 3g, 4b should be a comparison between SKMEL5+SKMEL26 and A2058+RPMI-7951 in the 1% FBS condition.

6. Figure 4b: Could the authors please add information on the biological and technical replicates included?

7. Figure 4c, d: It is very difficult to detect the suggested differences; I suggest including representative zoom-outs including a large number of cells.

8. Figure 4e, f:

a. PLIN2 and BODIPY C16 do not match well. Could the authors please explain this?

b. Could the authors please justify normalization to cell surface?

1b. The authors use a human cell line, A375, which does not present a melanocytic phenotype intrinsically. Instead, this had to be induced by IBMX or forskolin treatment (Figure 2C-E, 3C and 5D). Although a model like this may be used as supportive data, I wonder why the authors chose to not use their zebrafish system to study this question. Furthermore, the presence of the drugs could have an influence on the readout independently of the phenotypic change, which is not controlled for.

As noted above, we have added an additional n=4 human cell lines that are intrinsically either undifferentiated or melanocytic, allowing us to study this without IBMX/forskolin induction. In addition,

as noted below, we also used the zebrafish ZMEL1 melanoma line to test the effect of DGAT1 inhibition in vivo.

Reviewer 1 comments on rebuttal:

Comments on the additional cell lines are noted under major point 1a. In addition, the authors have added significant data supporting the in vivo and in vitro fitness effect of targeting DGAT1 with an inhibitor, showing a compelling in vivo effect.

1. Figure 6c, d, e:

a. Can the authors please confirm that the DGAT1 inhibitor effect is at least partially on target by performing the epistatic experiment? This could be done in vitro (figure 5e, adding a combined condition with the inhibitor and the guide).

b. Did the authors perform histology to exclude GFP loss as an explanation for the effect in figure 6d?

2. Figure 6e:

a. The in vitro validation of the DGAT1 inhibitor fitness effect should be a comparison between SKMEL5+SKMEL26 and A2058+RPMI-7951 in the 1% FBS condition.

b. There is great variation in the in vitro fitness experiment, especially for SKMEL28. Perhaps the authors could improve upon this using crystal violet, cell-titre blue or cell count read-outs to reduce the variation?

c. A2058 seems to behave more like a melanocytic cell line; can the authors comment on this?

1c. The upregulation of dct and pmel is not sufficient evidence that A375 is melanocytic (Figure 2D). To interpret these results the authors should add at least an additional cell line that is melanocytic to compare pmel & dct levels. In addition, determining the degree of melanin production may be informative.

We agree. As noted above, the data from Celligner allowed us to more comprehensively analyze pigmentation genes in TCGA and CCLE datasets outside of A375 alone and functionally test the additional n=4 representative cell lines. Examination of mita, dct, pmel and tyr all show higher expression (Supplementary Figure 5) in the melanocytic lines (i.e. SKMe15, SKMe128, amongst numerous others) vs undifferentiated lines (i.e. RPMI7951, A2058 amongst others).

Reviewer 1 comments on rebuttal:

For figure 2d: Can the authors add A375, A375+IBMX & A375 + Forskolin to panel 3f to confirm the phenotypes against the cell line panel?

1d. The authors use a seemingly confounding human pluripotent stem cell model (Figure 2A). They mention that seahorse experiments were performed with melanoblasts that were max. 7 days old. However, this did not seem to be the case for melanocytes derived from the stem cells, which were at least 100 days old. The difference in age of the cells could contribute to the differences in OCR described by the authors, for example if senescence/aging would occur.

We have previously demonstrated the utility of human pluripotent stem cells for modeling melanocytic differentiation from the neural crest (Baggiolini, Science 2021). The major advantage of this approach is that the cells are isogenic to each other, which is not the case when comparing across cell lines (i.e. CLE) or patient samples (i.e. TCGA), where baseline germline and somatic genetic background will differ. However, we acknowledge that a limitation of the iPS system is that the melanoblasts are isolated at a much earlier time point than melanocytes. This is an inherent limitation of differentiation in all stem cell models, since more differentiated progeny will always take longer to be generated than progenitors. Despite this, this system provides an orthogonal way to look at progenitor vs. melanocytic cells that complements what can be done in established cancer cell lines. At 100 days, the melanocytic cells are still actively proliferating and do not show evidence of senescence, making it unlikely this alone explains the OCR phenotypes. Moreover, the association between the melanocytic state and oxidative metabolism we see with the iPS cells is in line with what we now observe in the 5 cancer cell lines we have studied. We have added a consideration of this limitation to the manuscript text.

Reviewer 1 comments on rebuttal:

These changes are fine.

2: Novelty of the connection between the melanocytic cell state and oxidative phosphorylation. I am not entirely convinced that the association between the melanocytic cell state with oxidative phosphorylation is novel (Figure 1 and 2), as the authors acknowledge themselves in their Discussion. Similar findings have been described by Fischer, Haq, Gopal. For example, MITF, a key transcription factor of the melanocytic cell state, drives PGC1 α -mediated oxidative phosphorylation. Could the authors better explain what exactly is the new piece of data that advances our insight over those previous studies?

As you point out, the previous work has linked MITF to PGC1 α -mediated mitochondrial biogenesis to drive increased oxidative phosphorylation, but the substrates driving this and how the cell obtains those substrates is unknown. The goal of our study was to show that: 1) fatty acids are one key substrate, and that 2) this depends upon the uptake and processing in lipid droplets. While we cannot exclude that other substrates could be used, this data shows that fatty acids are especially important in the melanocytic state.

Reviewer 1 comments on rebuttal:

I appreciate the explanation made by the authors and recommend to add this to the discussion.

3: Questions about data bias. The authors performed RNA-seq on zebrafish tumors with sgNT or sgDGAT1a (Figure 5). This approach may have yielded interesting results, given the big clouds observed in the volcano plot. However, the biggest changes were not shown, let alone followed up. It looks like the authors instead (subjectively) chose to highlight several (marginally) affected genes in the data. This suggests data bias, which I am sure was not the authors' intention, but which should be addressed thoroughly. I recommend the authors uncover the biggest changes observed and take a more unbiased approach to better support their claims.

We agree this required a more clear and unbiased approach. In the new Figure 6, we show the top significantly enriched GO biological pathways, which includes those both up and downregulated. The data shown in this Figure are the pathways ordered by adjusted p-value. As shown, downregulation of cell division and cell cycle genes are amongst the top pathways. We also provide a full analysis of all differentially expressed genes in Supplementary Data 2. It was this effect on cell cycle that led us to hypothesize that DGAT1 deficiency would be associated with decreased proliferation.

To further test this, we have now measured the effect of a DGAT1 inhibitor both in vivo and in vitro. We transplanted ZMEL1 melanoma cells into zebrafish and treated them with the DGAT1 inhibitor T863 (Cayman Chemical Company, 25807) or DMSO control. This revealed a significant decrease in overall tumor burden (Figure 6d). To test if this was directly due to proliferation, we then tested the DGAT1 inhibitor in vitro in both the undifferentiated (RPMI7951, A2058) and melanocytic (SKMel5 and SKMel28) cell lines described above and measured proliferation using CyQuant. While the DGAT1 inhibitor had minimal effect in the undifferentiated cells, we saw a significant decrease in proliferation in the melanocytic lines (Figure 6e). Together, this new data is consistent with the RNA-seq showing a decrease in proliferation with DGAT loss or inhibition. While we agree there are many other interesting pathways uncovered in our RNA-seq, these will need to be further characterized in the future.

Reviewer 1 comments on rebuttal:

I agree with the changes made by the authors.

4: Level of support for the strongest claim. The authors repeatedly focus on the potential of DGAT1/lipid droplets as a therapeutic target, specifically in melanoma. This observation is highlighted in the title and is repeated throughout their discussion. Dependency on DGAT1/lipid droplets in melanoma specifically is an interesting observation. However, their actual data supporting this is limited to panel 4c, where the authors show reduced fitness of sgDGAT1a melanoma in their zebrafish model. This effect was unfortunately quite limited and not pursued or validated. This reduces the enthusiasm of this referee for the outcome of this study. Furthermore, the authors did not show lipid droplets by immunohistochemistry in zebrafish tumors for both the control and DGAT1a knock-out condition (Figure 4C and D).

We agree this needed stronger evidence. As mentioned above, we have now added in vitro and in vivo studies using a DGAT1 inhibitor. We find that the DGAT1 inhibitor inhibits proliferation in the melanocytic cell lines, and impairs tumor growth in zebrafish transplanted with the ZMEL1 melanoma cell line. We have also performed immunohistochemistry on the sgNT and sgDGAT1a tumors to probe for PLIN2. This is a protein found on the surface of the lipid droplet that we have previously shown specifically marks this organelle (Supplementary Figure 9).

Reviewer 1 comments on rebuttal:

I have already expressed my appreciation of the new zebrafish data above. Supplementary figure 9: Why is the BRAF staining intensity so much lower in fish #2?

Minor points:

1. The order of the phenotype labels is different in Figure 1E compared to 1B and 1C.

The order of clusters for Figure 1E is sorted in descending average melanocytic score. This is a conventional way of showing this for these analyses, although we can modify the order in categorical order according to b and c if the reviewer and editor prefer that.

Reviewer 1 comments on rebuttal:

I thank the authors for the explanation. For clarity, it would be helpful to have a consistent order between the figures; this can also be the conventional descending order in b, c and e.

2. In Figure 1B, 1C and 1E the authors present novel signatures, though they use the Tsoi signature for the melanocytic cell state. For consistency, it would be helpful to see whether the other 3 Tsoi signatures are also present in the single cell data.

Yes, we have found evidence of the other Tsoi signatures in our data, as shown below. As expected, the invasive state is most strongly associated with the undifferentiated or neural crest state.

Reviewer 1 comments on rebuttal:

I appreciate the analysis added by the authors; which could be added as a supplementary figure.

3. It is unclear why it is relevant to look at single cell data of metastatic brain melanoma. The authors should explain the choice for the Smalley dataset (Figure S3A).

This was largely a practicality. The Smalley dataset is the largest single cell data set in melanoma, and allowed us to validate our results of increased oxidative metabolic signatures in more melanocytic populations. In addition, human melanomas that metastasize to the brain have been shown to upregulate oxidative metabolism (Fischer, Cancer Discovery 2019) and drugs targeting this are currently being explored to target these metastases.

Reviewer 1 comments on rebuttal:

I appreciate the explanation by the authors. Given that brain melanomas upregulate oxidative metabolism, could the authors compare and contrast this to other single cell RNA sets like Tirosh et al., 2016 or Jerby-Arnon et al., 2018?

4. It is unclear why p53 was not deleted in the human pluripotent stem cell model (Figure 2a) for consistency with the zebrafish model (Figure 1a).

We used the iPS system so that we could have isogenic cells at different states (i.e.

melanoblast versus melanocyte) but did not focus on the role of p53 specifically here. We think it is unlikely that p53 explains the correlation between the melanocytic state and fatty acid oxidation, since it was not a consistent feature of either the n=5 cell lines we tested nor in the TCGA/CCLC Celligner analysis.

Reviewer 1 comments on rebuttal:

I appreciate the analysis and explanation by the authors. It would be helpful to know whether p53 loss is not a consistent feature in the 5 cell lines.

5. The fold change is small in Figure 2E (middle panel). One wonders whether such differences are biologically meaningful.

The magnitude of changes in Seahorse assays can often be subtle, since cells are typically not entirely oxidative or glycolytic. This is why the ratio of OCR/ECAR (reflecting oxidative vs glycolytic metabolism) are often more useful, since they show relative enrichment of one pathway over another.

Reviewer 1 comments on rebuttal:

In my original major point 1a I recommended the use of additional cell lines to validate figure 2e. I still think this is required to confirm the observation in the A375 model.

6. The normalization is different among Seahorse experiments presented (Figure 2B, 2E and 3a). This may affect or even confound the interpretation of these results.

Depending on the configuration of each assay plate and treatment (i.e. # of cells, density), it was sometimes more practical to normalize using BCA normalization (i.e. protein) and in other cases to normalize using nuclear fluorescence normalization. For each experiment, the conditions are only compared against each other and not across experiments. For example, Figures 2b, 2e and 3a are not compared against each other. This is why we indicated which normalization method was used for each experiment on the y-axis.

Reviewer 1 comments on rebuttal:

I agree with the explanation.

7. The left and right plots do not match (Figure 3A). The authors placed additional data in the supplement.

We had initially separated the Seahorse traces for ease of visualization, but agree this made the Figure confusing. We kept the key bar graph in Figure 3 but moved all of the Seahorse traces into Supplemental Figure 6 for consistency.

Reviewer 1 comments on rebuttal:

I agree with the changes. Supplemental figure 6b: axis should be consistent with 6a and 6c.

8. Starting points of the different conditions used in the Fatty acid uptake assay are different (Figure 3B). Also, one wonders whether 24 hours incubation of A375 with IBMX or forskolin can induce the melanocytic cell state. The authors showed in Figure 2 an incubation time of 3 days in order to reach this cell state. At the least, this seems inconsistent.

The slight increase in starting points is likely due to the fact that the forskolin and IBMX conditions have more lipid droplets at baseline (Supplementary Figure 7), which would increase the rate of uptake of dye. This is further evidenced by higher uptake over time. In terms of timing, prior studies have shown that forskolin can activate MITF driven programs within 24 hours (i.e. Khaled, Genes & Development 2010), but we agree that this may not be enough time to fully induce pigmentation. It is for those reasons that we chose to test additional cell lines, as mentioned above, representing the undifferentiated vs. melanocytic state (i.e. no need for forskolin or IBMX).

Reviewer 1 comments on rebuttal:

I am not yet convinced of the authors' explanation. In my view, all three conditions should be normalized to their own starting points. The logic behind this is that the uptake assay measures uptake over time; not the starting amount of lipid droplets. As it is, the difference at the start could be maintained throughout even if there is no increased uptake. I agree that more lipid droplets are present at the start according to supplementary figure 7.

9. The phenotype of the zebrafish is different between non targeting control and DGAT1a knockout condition. Furthermore, it seems that the intestine is more enlarged in the control condition (Figure 4D).

The original image was of a male and female fish, which is why they look so different. We have now replaced this with a new image that shows just the male fish for the sake of consistency (Figure 5b), but both male and female fish were included in the analysis.

Reviewer 1 comments on rebuttal:

I agree with the explanation and change.

10. The authors do not show that ZMEL-LD has a melanocytic phenotype (Figure 4A).

The ZMEL1 line has high levels of mitfa, and has been previously shown to rapidly pigment in both in vitro and in vivo conditions (Heilmann, Cancer Research 2015 and Kim, Nature Communications 2017).

Reviewer 1 comments on rebuttal:

I agree with the explanation.

11. The CRISPR knock-out does not cause RNA loss per se. Even if this is common, it is expected to be different for each guide (Figure 5B and in text comment on expectation).

We agree that this does not always happen with CRISPR. However, in this case, when we performed RNA-seq we found that there is a significant downregulation of dgat1a in the sgDGAT1a tumors. We have added the normalized counts in (Supplementary Figure 8).

Reviewer 1 comments on rebuttal:

I agree with the explanation and appreciate the added plot.

Reviewer #2 (Remarks to the Author):

I thank the authors for their detailed responses to reviewer critiques. I have no further concerns.

Reviewer #3 (Remarks to the Author):

Thanks to the authors for carefully considering my comments. The revisions have improved the manuscript, and I support publication.

Reviewer #4 (Remarks to the Author):

The authors have given me appropriate answers to my questions. I think it would be good to accept this manuscript.

REVIEWER COMMENTS

Reviewer #1 (Remarks to the Author):

Lumaquin et al made use of a BRAFV600/p53-/-/PTENko zebrafish melanoma model and single cell sequencing to describe different melanoma phenotypes. They identified a melanocytic phenotype characterized by increased oxidative phosphorylation and lipid droplet formation. The authors' main claim is that the melanocytic cell state is dependent on lipid droplets. Accordingly, they show that the oxidative metabolism is reduced upon suppression of lipid production. To translate their findings to in vivo, the authors perturbed lipid droplet formation by targeting DGAT1 using CRISPR technology. This resulted in (limited) tumor growth suppression in their melanoma zebrafish model.

Major points:

1. Limited cell line systems to support quite major claims.

1a. Although they start out in their well-known zebrafish model, the authors base many of their in vitro findings on a single human cell line model, A375 (Figure 2C-E, 3C and 5D). They could (should) have used a panel of cell lines that are phenotypically different to strengthen their claims. They could also mine databases to look for supportive data. For example, Tsoi et al. (2018) derived their different melanoma signatures from a panel of cell lines. Furthermore, the cancer cell line encyclopedia offers data (RNAseq, metabolomics, proteomics and more) in a large set of cell lines. Additionally, the authors could put together their own panel.

We have taken both a data mining approach as well as tested new representative cell lines. To identify cell lines and patient samples that better represented the entire spectrum of melanoma states, we utilized the data from the Celligner pipeline, which was optimized to integrate TCGA and CCLE datasets (Warren, Nature Communications 2021). This method allows for identification of gene signatures that are conserved across large scale human patient samples or cell lines. The integrated analysis (encompassing a total of n=38 cell lines and n=443 patient samples) revealed that melanomas fall into undifferentiated/neural crest, transitory and melanocytic states. In both the TCGA and CCLE datasets, the melanocytic cell state was associated with a significant enrichment for the fatty acid oxidation score (Figure 3a, Figure 3b and Supplementary Figure 5b,d). From the CCLE data, we then selected an additional n=4 cell lines for further functional analysis as you suggested. We chose 2 cell lines classified as undifferentiated/neural crest (RPMI7951 and A2058) and 2 classified as melanocytic (SKMel5 and SKMel28). As described below in more detail, we found that the melanocytic cell lines had increased fatty acid uptake (Figure 3g), DGAT1 mRNA (Figure 4a), triglyceride levels (Figure 4b) and lipid droplet area (Figure 4c-f).

Reviewer 1 comments on rebuttal:

I appreciate that the authors made the effort to provide considerable additional data supporting their in vitro findings. First of all, they added fatty acid oxidation gene scores from the CCLE and TCGA database which further supports their claims. Furthermore, the authors included several experiments in a panel of

cell lines with different phenotypes. Since such significant changes have been made to the figures and panels 3, 4 and s5 I have carefully evaluated the new data and listed my comments below.

1. Figure 3a, b: Is there overlap between the fatty acid oxidation gene set and the melanocytic gene set? If so, this could confound the correlation.

Out of the 110 genes in the fatty acid oxidation gene set and the 187 genes in the Tsoi melanocytic gene set, only 3 genes overlap: ABCD1, NR4A3 and PPARGC1A.

2. Figure 3e: The IBMX and Forskolin conditions already have an increase at time point zero, even though the authors note the data was normalized to time point zero.

We agree with this, and address it in point #8 below.

3. Figure 3f:

a. It would strengthen the data if the authors would include neural crest and undifferentiated markers.

These have been added to the Figure.

b. A2058 is not a compelling neural crest like cell line and I feel should be characterized as melanocytic transitory.

We agree, and in the manuscript text and figures we have correctly indicated this is a transitory line, but failed to state this properly in the prior rebuttal.

4. Figure 3g and figure s5: The authors start the y-axis at arbitrary numbers without clearly indicating this with the correct symbol (broken axis with two dashed lines). This should be corrected because it could lead to incorrect interpretation of the data.

For Figure 3g the dashed lines have been added, as requested. For Figure S5, it is customary to plot RNA-seq data in this manner since the starting place (i.e. normalized expression) varies from gene to gene, and so we have followed this convention.

5. Figure 3g & 4b: The validation of figure 3e in figure 3g, 4b should be a comparison between SKMEL5+SKMEL26 and A2058+RPMI-7951 in the 1% FBS condition.

We are cautious about comparing across different cell lines, because they are not isogenic to each other (i.e. there is substantial difference in both their germline and somatic DNA). This is why we think it is important to compare fatty acid uptake within a given cell line rather than across cell lines, since they will be isogenic and a more powerful control. The same logic applies to point #2 (regarding Fig. 6e and the DGAT1 inhibitor).

6. Figure 4b: Could the authors please add information on the biological and technical replicates included?

This has been added to the Figure legend and is also available in the source data.

7. Figure 4c, d: It is very difficult to detect the suggested differences; I suggest including representative zoom-outs including a large number of cells.

We have attempted the analysis you suggested, but because lipid droplets are relatively small, we found that the zoom-out images were not helpful in visualizing these differences. While the effects are subtle, this is expected given the small nature of lipid droplets, so feel the original image best represents what we have quantified.

8. Figure 4e, f:

a. PLIN2 and BODIPY C16 do not match well. Could the authors please explain this?

This is the expected result. PLIN2 is a lipid droplet coat protein, whereas BODIPY will stain fatty acids located inside of the lipid droplet. Thus while they will both localize to lipid droplets, they should have different staining patterns (i.e. outside versus inside).

b. Could the authors please justify normalization to cell surface?

We normalize to cell area to take into account that differences in overall area can lead to differential spatial capacity within the cytoplasm for forming and maintaining lipid droplets. This is an important control given that lipid droplets could differ based on overall available space.

1b. The authors use a human cell line, A375, which does not present a melanocytic phenotype intrinsically. Instead, this had to be induced by IBMX or forskolin treatment (Figure 2C-E, 3C and 5D). Although a model like this may be used as supportive data, I wonder why the authors chose to not use their zebrafish system to study this question. Furthermore, the presence of the drugs could have an influence on the readout independently of the phenotypic change, which is not controlled for.

As noted above, we have added an additional n=4 human cell lines that are intrinsically either undifferentiated or melanocytic, allowing us to study this without IBMX/forskolin induction. In addition, as noted below, we also used the zebrafish ZMEL1 melanoma line to test the effect of DGAT1 inhibition in vivo.

Reviewer 1 comments on rebuttal:

Comments on the additional cell lines are noted under major point 1a. In addition, the authors have added significant data supporting the in vivo and in vitro fitness effect of targeting DGAT1 with an inhibitor, showing a compelling in vivo effect.

1. Figure 6c, d, e:

a. Can the authors please confirm that the DGAT1 inhibitor effect is at least partially on target by performing the epistatic experiment? This could be done in vitro (figure 5e, adding a combined condition with the inhibitor and the guide).

T863 is a well characterized and widely used specific inhibitor of DGAT1, as described in the publication describing its identification (<https://pubmed.ncbi.nlm.nih.gov/21990351/>) and other publications have performed the epistasis experiments you have described (<https://pubmed.ncbi.nlm.nih.gov/35732120/>). Based on the literature and wide use of this compound, we do not think repeating the epistasis experiment, which is not trivial, would add to the manuscript.

b. Did the authors perform histology to exclude GFP loss as an explanation for the effect in figure 6d?

It is technically difficult to perform histology for fish in this developmental age. However, to exclude the possibility you raised, we have performed flow cytometry of the ZMEL-LD cells treated with the DGAT1 inhibitor and find there is no change in GFP expression, as shown below:

2. Figure 6e:

a. The in vitro validation of the DGAT1 inhibitor fitness effect should be a comparison between SKMEL5+SKMEL26 and A2058+RPMI-7951 in the 1% FBS condition.

As stated in point #5 above, we are cautious about comparing fitness effects between different cell lines, because they are not isogenic to each other (i.e. there is substantial difference in both their germline and somatic DNA). This is why we think it is important to compare the effect of the DGAT1 inhibitor to its own control, since they will be isogenic and a more powerful control.

b. There is great variation in the in vitro fitness experiment, especially for SKMEL28. Perhaps the authors could improve upon this using crystal violet, cell-titre blue or cell count read-outs to reduce the variation?

Cyquant is a direct readout of cell counts. While we agree that crystal violet or cell-titre blue are alternatives, we know of no data suggesting that these are superior to the direct cell counting afforded by Cyquant, and in our own hands are not less variable than Cyquant.

c. A2058 seems to behave more like a melanocytic cell line; can the authors comment on this?

As noted above, A2058 is transitory, meaning it possesses some melanocytic gene expression which could reflect sensitivity to DGAT1 inhibition.

1c. The upregulation of dct and pmel is not sufficient evidence that A375 is melanocytic (Figure 2D). To interpret these results the authors should add at least an additional cell line that is melanocytic to compare pmel & dct levels. In addition, determining the degree of melanin production may be informative.

We agree. As noted above, the data from Celligner allowed us to more comprehensively analyze pigmentation genes in TCGA and CCLE datasets outside of A375 alone and functionally test the additional n=4 representative cell lines. Examination of mita, dct, pmel and tyr all show higher expression (Supplementary Figure 5) in the melanocytic lines (i.e. SKMeI5, SKMeI28, amongst numerous others) vs undifferentiated lines (i.e. RPMI7951, A2058 amongst others).

Reviewer 1 comments on rebuttal:

For figure 2d: Can the authors add A375, A375+IBMX & A375 + Forskolin to panel 3f to confirm the phenotypes against the cell line panel?

We have not performed RNA-seq on the A375 cells treated with IBMX or forskolin so do not think it would be appropriate to add to the heatmap in Figure 3f (which was derived from the CCLE dataset).

1d. The authors use a seemingly confounding human pluripotent stem cell model (Figure 2A). They mention that seahorse experiments were performed with melanoblasts that were max. 7 days old. However, this did not seem to be the case for melanocytes derived from the stem cells, which were at least 100 days old. The difference in age of the cells could contribute to the differences in OCR described by the authors, for example if senescence/aging would occur.

We have previously demonstrated the utility of human pluripotent stem cells for modeling melanocytic differentiation from the neural crest (Baggiolini, Science 2021). The major advantage of this approach is that the cells are isogenic to each other, which is not the case when comparing across cell lines (i.e. CCL) or patient samples (i.e. TCGA), where baseline germline and somatic genetic background will differ. However, we acknowledge that a limitation of the iPS system is that the melanoblasts are isolated at a much earlier time point than melanocytes. This is an inherent limitation of differentiation in all stem cell models, since more differentiated progeny will always take longer to be generated than progenitors. Despite this, this system provides an orthogonal way to look at progenitor vs. melanocytic cells that complements what can be done in established cancer cell lines. At 100 days, the melanocytic cells are still actively proliferating and do not show evidence of senescence, making it unlikely this alone explains the OCR phenotypes. Moreover, the association between the melanocytic state and oxidative metabolism we see with the iPS cells is in line with what we now observe in the 5 cancer cell lines we have studied. We have added a consideration of this limitation to the manuscript text.

Reviewer 1 comments on rebuttal:

These changes are fine.

2: Novelty of the connection between the melanocytic cell state and oxidative phosphorylation. I am not entirely convinced that the association between the melanocytic cell state with oxidative phosphorylation is novel (Figure 1 and 2), as the authors acknowledge themselves in their Discussion. Similar findings have been described by Fischer, Haq, Gopal. For example, MITF, a key transcription factor of the melanocytic cell state, drives PGC1 α -mediated oxidative phosphorylation. Could the authors better explain what exactly is the new piece of data that advances our insight over those previous studies?

As you point out, the previous work has linked MITF to PGC1 α -mediated mitochondrial biogenesis to drive increased oxidative phosphorylation, but the substrates driving this and how the cell obtains those substrates is unknown. The goal of our study was to show that: 1) fatty acids are one key substrate, and that 2) this depends upon the uptake and processing in lipid droplets. While we cannot exclude that other substrates could be used, this data shows that fatty acids are especially important in the melanocytic state.

Reviewer 1 comments on rebuttal:

I appreciate the explanation made by the authors and recommend to add this to the discussion.

This has been added.

3: Questions about data bias. The authors performed RNA-seq on zebrafish tumors with sgNT or sgDGAT1a (Figure 5). This approach may have yielded interesting results, given the big clouds observed in the volcano plot. However, the biggest changes were not shown, let alone followed up. It looks like the authors instead (subjectively) chose to highlight several (marginally) affected genes in the data. This suggests data bias, which I am sure was not the authors' intention, but which should be addressed thoroughly. I recommend the authors uncover the biggest changes observed and take a more unbiased approach to better support their claims.

We agree this required a more clear and unbiased approach. In the new Figure 6, we show the top significantly enriched go biological pathways, which includes those both up and downregulated. The data shown in this Figure are the pathways ordered by adjusted p-value. As shown, downregulation of cell division and cell cycle genes are amongst the top pathways. We also provide a full analysis of all differentially expressed genes in Supplementary Data 2. It was this effect on cell cycle that led us to hypothesize that DGAT1 deficiency would be associated with decreased proliferation.

To further test this, we have now measured the effect of a DGAT1 inhibitor both in vivo and in vitro. We transplanted ZMEL1 melanoma cells into zebrafish and treated them with the DGAT1 inhibitor T863 (Cayman Chemical Company, 25807) or DMSO control. This revealed a significant decrease in overall tumor burden (Figure 6d). To test if this was directly due to proliferation, we then tested the DGAT1 inhibitor in vitro in both the undifferentiated (RPMI7951, A2058) and melanocytic (SKMeI5 and SKMeI28) cell lines described above and measured proliferation using CyQuant. While the DGAT1 inhibitor had minimal effect in the undifferentiated cells, we saw a significant decrease in proliferation in the melanocytic lines (Figure 6e). Together, this new data is consistent with the RNA-seq showing a decrease in proliferation with DGAT loss or inhibition. While we agree there are many other interesting pathways uncovered in our RNA-seq, these will need to be further characterized in the future.

Reviewer 1 comments on rebuttal:

I agree with the changes made by the authors.

4: Level of support for the strongest claim. The authors repeatedly focus on the potential of DGAT1/lipid droplets as a therapeutic target, specifically in melanoma. This observation is highlighted in the title and is repeated throughout their discussion. Dependency on DGAT1/lipid droplets in melanoma specifically is an interesting observation. However, their actual data supporting this is limited to panel 4c, where the authors show reduced fitness of sgDGAT1a melanoma in their zebrafish model. This effect was unfortunately quite limited and not pursued or validated. This reduces the enthusiasm of this referee for the outcome of this study. Furthermore, the authors did not show lipid droplets by immunohistochemistry in zebrafish tumors for both the control and DGAT1a knock-out condition (Figure 4C and D).

We agree this needed stronger evidence. As mentioned above, we have now added in vitro and in vivo studies using a DGAT1 inhibitor. We find that the DGAT1 inhibitor inhibits proliferation in the melanocytic cell lines, and impairs tumor growth in zebrafish transplanted with the ZMEL1 melanoma cell line. We

have also performed immunohistochemistry on the sgNT and sgDGAT1a tumors to probe for PLIN2. This is a protein found on the surface of the lipid droplet that we have previously shown specifically marks this organelle (Supplementary Figure 9).

Reviewer 1 comments on rebuttal:

I have already expressed my appreciation of the new zebrafish data above. Supplementary figure 9: Why is the BRAF staining intensity so much lower in fish #2?

In transgenic animals, variable expression of mutant BRAFV600E is commonly seen, as is the case for human patients. The reasons for this are poorly understood. But in our own single-cell RNA data, it can readily be seen that transcript levels of BRAF vary substantially, suggesting that at least one mechanism may be transcriptional. A plot demonstrating this is below.:

Minor points:

1. The order of the phenotype labels is different in Figure 1E compared to 1B and 1C.

The order of clusters for Figure 1E is sorted in descending average melanocytic score. This is a conventional way of showing this for these analyses, although we can modify the order in categorical order according to b and c if the reviewer and editor prefer that.

Reviewer 1 comments on rebuttal:

I thank the authors for the explanation. For clarity, it would be helpful to have a consistent order between the figures; this can also be the conventional descending order in b, c and e.

As is conventional in RNA-seq, we show the plots in descending order based on the UMAP and Seurat algorithm (b and c) or the Tsoi melanocytic program score from the original publication (e). We think this best reflects the convention in the field to demonstrate data in descending order.

2. In Figure 1B, 1C and 1E the authors present novel signatures, though they use the Tsoi signature for the melanocytic cell state. For consistency, it would be helpful to see whether the other 3 Tsoi signatures are also present in the single cell data.

Yes, we have found evidence of the other Tsoi signatures in our data, as shown below. As expected, the invasive state is most strongly associated with the undifferentiated or neural crest state.

Reviewer 1 comments on rebuttal:

I appreciate the analysis added by the authors; which could be added as a supplementary figure.

These have been added.

3. It is unclear why it is relevant to look at single cell data of metastatic brain melanoma. The authors should explain the choice for the Smalley dataset (Figure S3A).

This was largely a practicality. The Smalley dataset is the largest single cell data set in melanoma, and allowed us to validate our results of increased oxidative metabolic signatures in more melanocytic populations. In addition, human melanomas that metastasize to the brain have been shown to upregulate oxidative metabolism (Fischer, Cancer Discovery 2019) and drugs targeting this are currently being explored to target these metastases.

Reviewer 1 comments on rebuttal:

I appreciate the explanation by the authors. Given that brain melanomas upregulate oxidative metabolism, could the authors compare and contrast this to other single cell RNA sets like Tirosh et al., 2016 or Jerby-Arnon et al., 2018?

We have done the analysis you suggested but do not see as clear an enrichment in these other data sets. For Tirosh, this likely reflects the relatively small number of cells in that dataset. For Jerby-Arnon, this likely reflects the high variation in biopsy sites used in their Cohorts 1&2, which may not be equally enriched for the melanocytic program as has been previously seen for brain metastases.

4. It is unclear why p53 was not deleted in the human pluripotent stem cell model (Figure 2a) for consistency with the zebrafish model (Figure 1a).

We used the iPS system so that we could have isogenic cells at different states (i.e. melanoblast versus melanocyte) but did not focus on the role of p53 specifically here. We think it is unlikely that p53 explains the correlation between the melanocytic state and fatty acid oxidation, since it

was not a consistent feature of either the n=5 cell lines we tested nor in the TCGA/CCLL Celligner analysis.

Reviewer 1 comments on rebuttal:

I appreciate the analysis and explanation by the authors. It would be helpful to know whether p53 loss is not a consistent feature in the 5 cell lines.

This is not a consistent feature, as noted in the table below:

Cell Line	TP53
A375	WT
RPMI-7951	Nonsense (pS166)
A2058	Missense (pV274F)
SKMEL5	WT
SKMEL28	Missense (pL145R)

5. The fold change is small in Figure 2E (middle panel). One wonders whether such differences are biologically meaningful.

The magnitude of changes in Seahorse assays can often be subtle, since cells are typically not entirely oxidative or glycolytic. This is why the ratio of OCR/ECAR (reflecting oxidative vs glycolytic metabolism) are often more useful, since they show relative enrichment of one pathway over another.

Reviewer 1 comments on rebuttal:

In my original major point 1a I recommended the use of additional cell lines to validate figure 2e. I still think this is required to confirm the observation in the A375 model.

In Supplementary Figure 10, using Seahorse we show that there are greater effects on oxidative metabolism in the melanocytic cell lines treated with the DGAT1 inhibitor. Given the caveats inherent to the IBMX/forskolin experiments, and the very time consuming nature of repeating these experiments in more cell lines, we feel this other data is a better orthogonal validation of the effects we describe, and consistent with the other data presented in the paper.

6. The normalization is different among Seahorse experiments presented (Figure 2B, 2E and 3a). This may affect or even confound the interpretation of these results.

Depending on the configuration of each assay plate and treatment (i.e. # of cells, density), it was sometimes more practical to normalize using BCA normalization (i.e. protein) and in other cases to normalize using nuclear fluorescence normalization. For each experiment, the conditions are only compared against each other and not across experiments. For example, Figures 2b, 2e and 3a are not compared against each other. This is why we indicated which normalization method was used for each experiment on the y-axis.

Reviewer 1 comments on rebuttal:

I agree with the explanation.

7. The left and right plots do not match (Figure 3A). The authors placed additional data in the supplement.

We had initially separated the Seahorse traces for ease of visualization, but agree this made the Figure confusing. We kept the key bar graph in Figure 3 but moved all of the Seahorse traces into Supplemental Figure 6 for consistency.

Reviewer 1 comments on rebuttal:

I agree with the changes. Supplemental figure 6b: axis should be consistent with 6a and 6c.

This has been changed as requested.

8. Starting points of the different conditions used in the Fatty acid uptake assay are different (Figure 3B). Also, one wonders whether 24 hours incubation of A375 with IBMX or forskolin can induce the melanocytic cell state. The authors showed in Figure 2 an incubation time of 3 days in order to reach this cell state. At the least, this seems inconsistent.

The slight increase in starting points is likely due to the fact that the forskolin and IBMX conditions have more lipid droplets at baseline (Supplementary Figure 7), which would increase the rate of uptake of dye. This is further evidenced by higher uptake over time. In terms of timing, prior studies have shown that forskolin can activate MITF driven programs within 24 hours (i.e. Khaled, Genes & Development 2010), but we agree that this may not be enough time to fully induce pigmentation. It is for those reasons that we chose to test additional cell lines, as mentioned above, representing the undifferentiated vs. melanocytic state (i.e. no need for forskolin or IBMX).

Reviewer 1 comments on rebuttal:

I am not yet convinced of the authors' explanation. In my view, all three conditions should be normalized to their own starting points. The logic behind this is that the uptake assay measures uptake over time; not the starting amount of lipid droplets. As it is, the difference at the start could be maintained throughout even if there is no increased uptake. I agree that more lipid droplets are present at the start according to supplementary figure 7.

We now see your point more clearly and agree that plotting the data this way could be confounding the interpretation. We have performed the normalization as you requested to each cell line's own starting

point, and indeed see that the majority of the effect is due to the starting condition (i.e. more lipid droplets to begin with, as shown in Supplementary Figure 7) that is then maintained over time. This is demonstrated in the plot below:

Based on this, we feel that it is most appropriate to reword the conclusion from Figure 3e to state that IBMX and Forskolin increase the number of lipid droplets, but that they do not increase the rate of lipid uptake above and beyond this at later time points. This important clarification has been added to the manuscript text: *“This revealed that the more melanocytic cells had an increased number of lipid droplets (Supplementary Figure 7), and a corresponding increase in lipid uptake compared to control A375 cells (Fig. 3e). However, they do not accelerate their uptake over time, suggesting that the number of lipid droplets induced by IBMX/forskolin at that starting point (i.e. time 0 in Figure 3e) is the main determinant of overall lipid uptake in these conditions.*

9. The phenotype of the zebrafish is different between non targeting control and DGAT1a knockout condition. Furthermore, it seems that the intestine is more enlarged in the control condition (Figure 4D). The original image was of a male and female fish, which is why they look so different. We have now replaced this with a new image that shows just the male fish for the sake of consistency (Figure 5b), but both male and female fish were included in the analysis.

Reviewer 1 comments on rebuttal:

I agree with the explanation and change.

10. The authors do not show that ZMEL-LD has a melanocytic phenotype (Figure 4A).

The ZMEL1 line has high levels of *mitfa*, and has been previously shown to rapidly pigment in both in vitro and in vivo conditions (Heilmann, Cancer Research 2015 and Kim, Nature Communications 2017).

Reviewer 1 comments on rebuttal:

I agree with the explanation.

11. The CRISPR knock-out does not cause RNA loss per se. Even if this is common, it is expected to be different for each guide (Figure 5B and in text comment on expectation).

We agree that this does not always happen with CRISPR. However, in this case, when we performed RNA-seq we found that there is a significant downregulation of *dgat1a* in the *sgDGAT1a* tumors. We have added the normalized counts in (Supplementary Figure 8).

Reviewer 1 comments on rebuttal:

I agree with the explanation and appreciate the added plot.

Reviewer #2 (Remarks to the Author):

I thank the authors for their detailed responses to reviewer critiques. I have no further concerns.

Reviewer #3 (Remarks to the Author):

Thanks to the authors for carefully considering my comments. The revisions have improved the manuscript, and I support publication.

Reviewer #4 (Remarks to the Author):

The authors have given me appropriate answers to my questions. I think it would be good to accept this manuscript.

REVIEWERS' COMMENTS

Reviewer #1 (Remarks to the Author):

The authors have resolved most of my comments and the paper is acceptable for publication.

Some final remarks:

- 1) As far as I could see, epistasis experiments were not performed in <https://pubmed.ncbi.nlm.nih.gov/35732120/>, only comparisons between drug treatment and knockdowns. However, several different compounds show the same effect in that paper.
- 2) Figure 6c, d, e: I am happy that upon my repeated request the authors have performed a reanalysis of the data and now indeed conclude that plotting the data this way was confounding the interpretation, and that the conclusion was accordingly adjusted.
- 3) The authors should consider including the comment "For Tirosh, this likely reflects the relatively small number of cells in that dataset. For Jerby-Anon, this likely reflects the high variation in biopsy sites used in their Cohorts 1&2, which may not be equally enriched for the melanocytic program as has been previously seen for brain metastases."

REVIEWER COMMENTS

Reviewer #1 (Remarks to the Author):

The authors should consider including the comment “For Tirosh, this likely reflects the relatively small number of cells in that dataset. For Jerby-Anon, this likely reflects the high variation in biopsy sites used in their Cohorts 1&2, which may not be equally enriched for the melanocytic program as has been previously seen for brain metastases.”

A consideration of this has been added to the manuscript.